# Chain flexibility of medicinal lipids determines their selective partitioning into lipid droplets

So-Hee Son[1,2], Gyuri Park[3], Junho Lim[4], Chang Yun Son ● [4,5,6✉], Seung Soo Oh ● [2,3,6✉] & Ju Young Lee ● [1✉]

In guiding lipid droplets (LDs) to serve as storage vessels that insulate high-value lipophilic compounds in cells, we demonstrate that chain flexibility of lipids determines their selective migration in intracellular LDs. Focusing on commercially important medicinal lipids with biogenetic similarity but structural dissimilarity, we computationally and experimentally validate that LD remodeling should be differentiated between overproduction of structurally flexible squalene and that of rigid zeaxanthin and β-carotene. In molecular dynamics simulations, worm-like flexible squalene is readily deformed to move through intertwined chains of triacylglycerols in the LD core, whereas rod-like rigid zeaxanthin is trapped on the LD surface due to a high free energy barrier in diffusion. By designing yeast cells with either much larger LDs or with a greater number of LDs, we observe that intracellular storage of squalene significantly increases with LD volume expansion, but that of zeaxanthin and β-carotene is enhanced through LD surface broadening; as visually evidenced, the outcomes represent internal penetration of squalene and surface localization of zeaxanthin and β-carotene. Our study shows the computational and experimental validation of selective lipid migration into a phase-separated organelle and reveals LD dynamics and functionalization.

[1] Research Center for Bio-based Chemistry, Korea Research Institute of Chemical Technology (KRICT), 406-30, Jongga-ro, Jung-gu, Ulsan 44429, Republic of Korea. [2] School of Interdisciplinary Bioscience and Bioengineering, Pohang University of Science and Technology (POSTECH), Pohang 37673, Republic of Korea. [3] Department of Materials Science and Engineering, Pohang University of Science and Technology (POSTECH), Pohang 37673, Republic of Korea. [4] Department of Chemistry, Pohang University of Science and Technology (POSTECH), Pohang 37673, Republic of Korea. [5] Division of Advanced Materials Science, Pohang University of Science and Technology (POSTECH), Pohang 37673, Republic of Korea. [6] Institute for Convergence Research and Education in Advanced Technology (I-CREATE), Yonsei University, Incheon 21983, Republic of Korea. ✉email: changyunson@postech.ac.kr; seungsoo@postech.ac.kr; juylee@krict.re.kr

A lipid droplet (LD) is not a simple blob of fat but a highly dynamic organelle capable of regulating lipid metabolism, storage, and transportation[1–5]. In particular, the phase-separated organelle can store intracellularly produced lipids with reservation, ensuring cell survival and stress resistance by promoting organelle and membrane homeostasis[6–8]. Under starvation conditions, the LD allows triacylglycerols (TAG) to be hydrolyzed into free fatty acids (FAs) and glycerol for energy production by mitochondrial FA oxidation[9]. When cellular membranes are in high demand, the TAG hydrolysis can provide lipids for phospholipid synthesis and maintenance[10,11]. However, under nutrient surplus conditions, excess FAs and sterols are stored in the LD as energy-rich TAGs and cholesteryl esters[6,12]. Importantly, as the FAs often disrupt membrane integrity affecting its permeability, they can be sequestered into the LD to protect a cell against lipotoxicity and oxidative stress[13,14]. This FA buffering strategy is highly valuable for engineering applications; for example, intracellular LDs can be remodeled to serve as excellent reservoirs that insulate lipophilic natural products, when overproduced for use in pharmaceuticals and nutraceuticals, thereby maximizing cell sustainability and metabolic efficiency[15,16].

For lipid metabolism and storage, molecules are transported in and out of a LD by crossing its phospholipid monolayer. Typically, membrane-bound organelles (e.g., the nucleus, mitochondria, the endoplasmic reticulum (ER), and the Golgi apparatus) are enveloped by a phospholipid bilayer; these membranes are impermeable to hydrophilic molecules, but hydrophobic lipids are trapped in the bilayer, diffusing laterally through the phospholipid matrix[17]. In contrast to bilayer-bound organelles, the TAG-encapsulating LD relies on a monolayer membrane, wherein the hydrophobic tails of phospholipids point toward the hydrophobic LD core[6]. This structure indicates that lipophilic lipids readily pass through the ultrathin membrane and migrate into the condensed lipophilic TAGs. However, experimental findings did not support this hydrophobicity-based transportation hypothesis[18]; the limited storage of significantly hydrophobic lycopene in spacious LDs was recently reported[19], and surprisingly, some fat-soluble drugs were selectively partitioned into LDs, highlighting the role of LDs in lipid-dependent storage and transportation[15]. Despite the explosion of interest in LD dynamics and functions, an in-depth understanding of the LD's lipid partitioning is currently lacking, undermining the potential for its many applications in synthetic biology, including metabolite reservoir remodeling[20].

In this study, by combining molecular dynamics (MD) simulations and microbial LD engineering, we demonstrate that the structural flexibility of lipids is one of decisive factors in selective diffusion and consequent partitioning of lipids into cytosolic LDs (Fig. 1). Within a cell, intracellularly produced lipids are easily captured by hydrophobic interactions on the phospholipid membrane of LDs (Fig. 1A, Interface I). However, for further migration, these captured lipids must pass through tightly packed TAGs (Interface II). In MD simulations, flexible lipids are readily deformed to move through intertwined chains of TAGs, while the diffusion of rigid ones causes a severe geometric rearrangement of the TAG chains, and these lipids are retained, unable to enter the LD core. As we have developed an LD engineering strategy enabling control over the total volume and net surface area of intracellular LDs, we are able to investigate the storage dependency of non-saponifiable lipids in yeast: squalene with isolated π bonds (Fig. 1B, blue) and zeaxanthin with conjugated π bonds (red). Representing a class of commercially important terpenes that cannot be sufficiently synthesized to meet market demands by microbial cells[21,22], they are successfully harbored in our remodeled LDs; intracellular storage of squalene is dramatically increased, by ~3100%, with LD volume expansion, and that of zeaxanthin is enhanced through LD surface broadening. This study offers computational and experimental validation of the chain flexibility-dependent, selective migration of lipids into LDs, providing insights for LD dynamics and functionalization applications.

## Results

**MD simulations of selective lipid migration into LDs**. As flexible FAs are known to freely enter or exit a cytosolic LD in lipid metabolism[6], we hypothesized that the structural flexibility of lipids influences their transportation through the phospholipids surrounding LDs. To test this hypothesis, we used all-atom MD simulations to compare cytosol-to-LD migration of two lipophilic molecules with biogenetic similarity but structural dissimilarity (i.e., flexible squalene and rigid zeaxanthin) as they can be intracellularly produced by our engineered cells[21]. In simulating the phase-separated organelle, our LD model distinctively established two different boundaries: one between the cytosol and phospholipid monolayer (Fig. 1A, Interface I) and the other between the phospholipids and TAGs (Interface II). By performing unrestrained MD simulations and calculating the free energy landscape using metadynamics, we compared the migration process of the flexible and rigid lipids passing through the two distinct boundaries (see Methods).

Boundary-dependent, two-step lipid migration was observed in the unrestrained, extensive 500-ns-long all-atom MD simulations (Fig. 2, and Supplementary Movie 1 and 2). At Interface I, both squalene and zeaxanthin displayed fast penetration from the cytosol into the phospholipid monolayer (<10 ns for both squalene and zeaxanthin, as shown in Fig. 2A, B, respectively, and Fig. 2C), mainly due to hydrophobic interactions; as lipid-solvating water molecules were removed, the hydrophobic lipids were immediately embedded into the phospholipid monolayer. However, at Interface II, the two lipids exhibited drastically different behaviors (within 50~70 ns). While conformationally changeable squalene continued to penetrate the TAG matrix, the structurally rigid zeaxanthin was trapped in the narrow boundary between the phospholipid tails and the TAGs in the LD core. During 500-ns simulations, worm-like squalene successfully migrated deep into the TAG-enriched LD center (Fig. 2C, blue), but rod-like zeaxanthin failed to pass the phospholipid-TAG boundary, and stalled at Interface II (red), showing slow orientation angle ($\theta$) fluctuation (Supplementary Fig. 1).

Such a striking difference in the lipid migration dynamics across the ultrathin phospholipid monolayer is the result of different conformations of the aliphatic tails in phospholipids and TAGs: While the paired tails of the phospholipids are aligned in parallel in the monolayer, the three FA tails of TAGs are randomly distributed, forming intertwined chains in a polymer melt-like state[23]. Indeed, during the entire migration process, squalene and zeaxanthin underwent completely different structural changes, regardless of the LD location. When the end-to-end distance of lipids, $R_{\text{end-to-end}}$, was quantified (Fig. 2D), the length of squalene (blue line) randomly changed within a range from 0.5 to 2.4 nm with great conformational fluctuations, but that of zeaxanthin (red line) remained constant (~2.4 nm). From this observation of the $R_{\text{end-to-end}}$ data, we expected flexible squalene to diffuse through the small interstitial free areas among tightly packed TAGs because its shape and orientation consistently adjusts. On the other hand, rigid zeaxanthin encountered a strong free energy barrier upon entering the TAG matrix, as geometric rearrangement of the intertwined TAG chains was required to create vacancies for accommodating unfoldable lipids. To identify this lipid-dependent diffusion barrier, we measured the solute-dependent friction

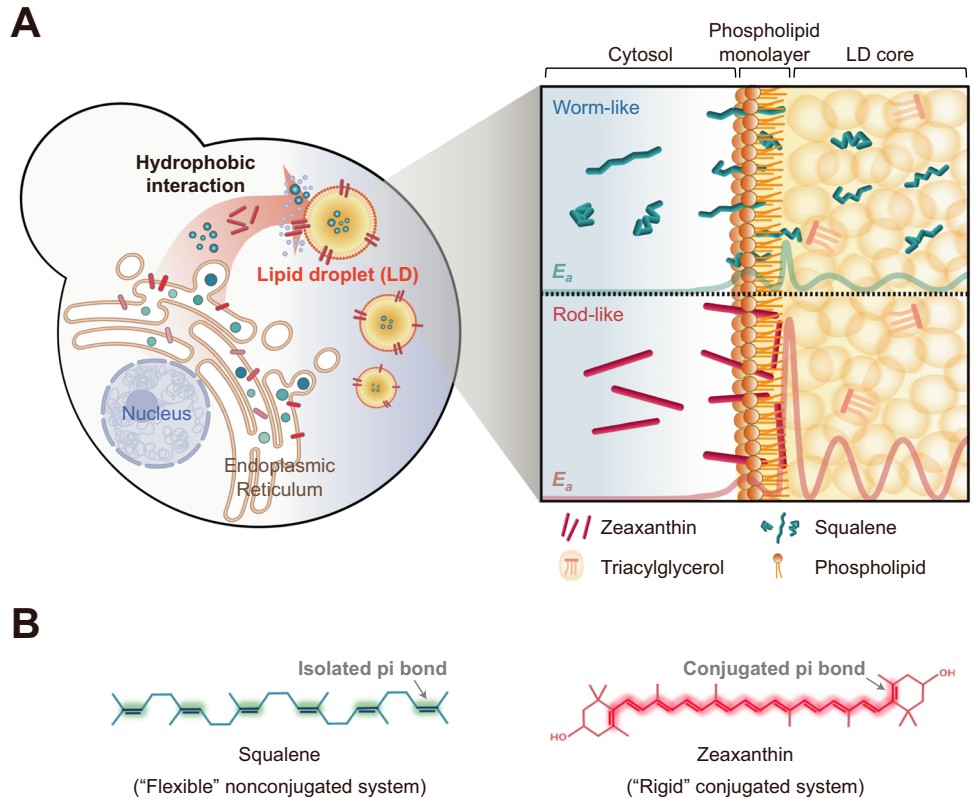

**Fig. 1 Chain flexibility-dependent selective migration of lipids into lipid droplets (LDs).** **A** Migration process of worm-like and rod-like lipids across cytosol-to-phospholipid and phospholipid-to-triacylglycerol (TAG) boundaries (Interfaces I and II, respectively). When lipids are intracellularly produced in living organisms such as yeast, cytosolic LDs, as phase-separated organelles, can attract lipophilic molecules through strong hydrophobic interactions, allowing their subsequent penetration through Interface I. However, to migrate deep into LDs, lipids must move through tightly packed TAGs at Interface II. Flexible lipids (blue) are readily deformed, lowering the free energy barrier, and pass through the tightly packed TAGs. However, to migrate deep into a LD, rigid lipids (red) must drastically rearrange intertwined TAG chains, and due to the elevated free energy barrier, unfoldable lipids are stalled in the membrane and cannot enter the LD core. **B** Chemical structure of the non-saponifiable lipids used in our computational and experimental analyses. Because of isolated π bonds, squalene (blue) is structurally flexible, but zeaxanthin (red), with conjugated π bonds, is rigid, exhibiting negligible conformational changes.

coefficient in equilibrated neat TAGs by steered MD simulations[24] (Fig. 2E). When each lipid solute molecule was pulled with a constant external force (200 kJ/mol/nm), calculations of friction coefficient ζ by the Langevin equation revealed that frictional forces against worm-like squalene are twofold weaker than those against rod-like zeaxanthin, effectively describing the reason that only squalene migrates deep inside TAGs.

By performing multiwalker well-tempered metadynamics simulations[25], we quantitatively calculated the free energy barrier of lipid migration into LDs through free energy landscape visualization (as shown in Fig. 2F, G for squalene and zeaxanthin, respectively). Across Interface I (z ~ 0), both squalene and zeaxanthin displayed a profound decrease in free energy (>12 kcal/mol) upon moving from the cytosol into the phospholipid layer, which was consistent in their rapid adsorption and penetration in the bias-free MD simulations (t ~ 0 in Fig. 2C). At Interface II (z ~ −2.76 nm), the free energy barriers of squalene and zeaxanthin were ~12 and 9 kcal/mol, respectively, proving that complete squalene migration into LDs was more likely than zeaxanthin migration in LDs. Moreover, as the two lipids presented different conformations with preferred two-order parameter ranges (migration depth z and conformation order parameter μ, see Methods), the free energy landscape of squalene was more diffuse than that of zeaxanthin. Specifically, across Interface II, worm-like squalene underwent conformational changes to reach multiple local free energy minima (blue, Fig. 2F),

but rod-like zeaxanthin displayed free rotational motion only when trapped at the phospholipid-TAG boundary region (−2.2 ≤ z ≤ −1.5, Fig. 2G). Importantly, from the perspective of orientational dynamics, a plausible pathway for selective lipid migration was identified (arrows in Fig. 2F), which started from a collapsed conformation on the phospholipid side of Interface II (at z ~ −2 and μ ≤ 0.2) and underwent a transition state of moderate-sized bent conformation (at z ~ −2.5 and μ ~ 0.4) before reaching the local free energy minima within the LD core (blue).

**Microbial control over the size and number of intracellular LDs.** Given the MD prediction, in vivo validation of the structural flexibility-dependent lipid transportation was necessary, and we thus devised a LD engineering strategy capable of controlling the size and number of cytosolic LDs within a living cell. When lipids are preferentially transported and located in different areas of LDs, the storage within the LD is directly influenced; that is, the storage capacity of target lipids is proportional to the total volume or the net surface area of the core-shell structure. Specifically, as flexible lipids are presumably stored inside the TAG core of the LD, LD size expansion causes a dramatic increase in the intracellular storage of foldable lipids. On the other hand, rigid lipids are preferentially retained on the LD surface, which was evaluated with our engineering strategy for increasing the number of LDs.

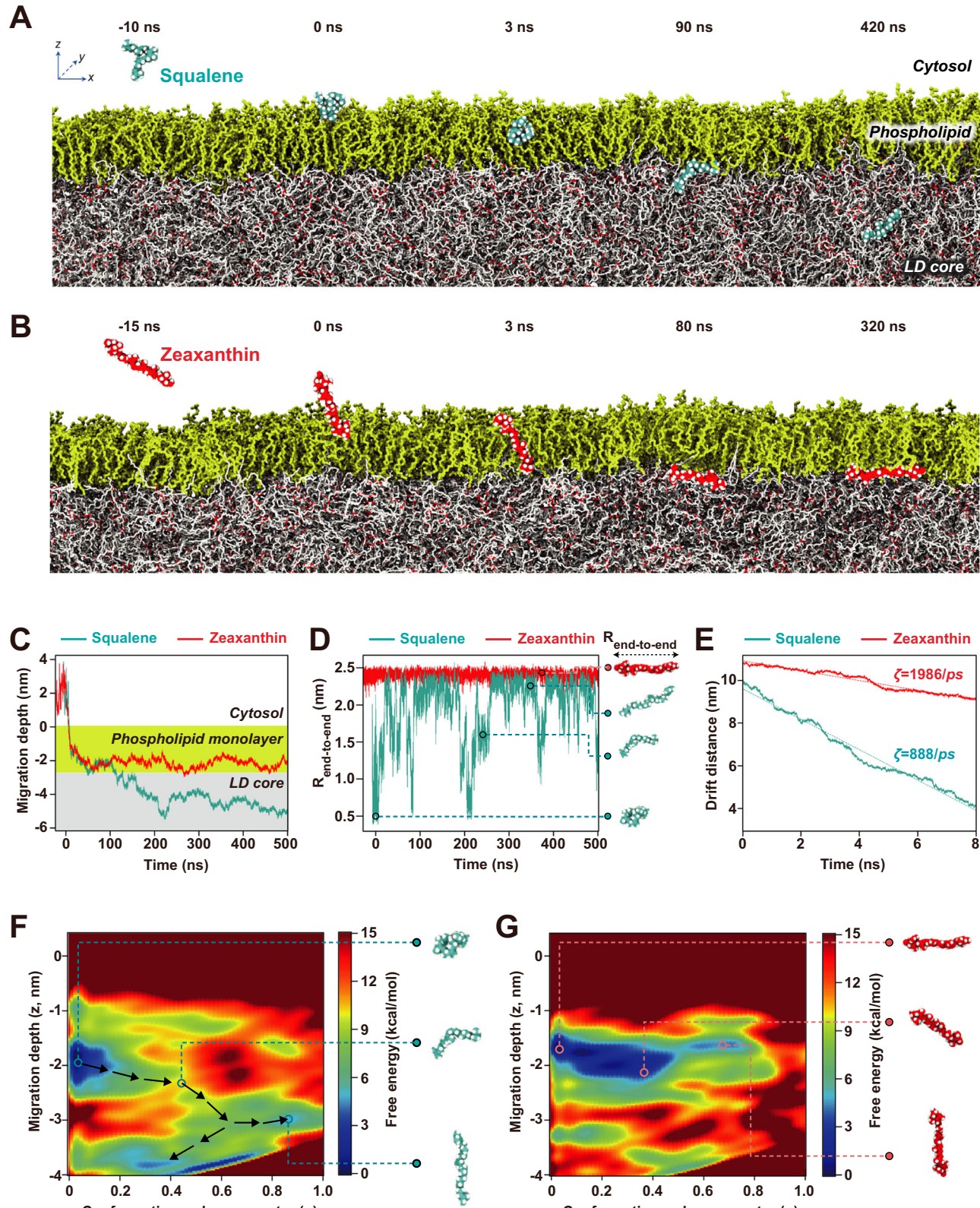

Notably, dynamic living cells did not allow us to fully control the net surface area of intracellular LDs; however, we assessed the effect of the lipid localization on the monolayer surface by increasing the number of spherical LDs, each with a diameter similar to that of wild-type (WT) LDs.

Then, by systematically regulating LD biogenesis and degradation pathways, we readily controlled the size and number of

intracellular LDs. As an exemplar model, a yeast cell was engineered for our experiments because various genes are known to be relevant to saccharomyces LDs. In particular, for our LD size and number engineering strategy, we focused on ten different genes associated with physicochemically distinct functions: (i) *lro1* and *dga1* for TAG synthesis, (ii) *tgl3*, *tgl4*, and *tgl5* for TAG degradation, (iii) *cld1* and *pex10* for TAG precursor supply, and

**Fig. 2 MD simulations of lipid migration across the cytosol-to-phospholipid and phospholipid-to-TAG boundaries of LDs. A, B** Representative snapshots from bias-free MD simulations are visualized for (**A**) flexible worm-like squalene and (**B**) rigid rod-like zeaxanthin. The phospholipid monolayer is aligned in the xy-plane, and when initially placed in aqueous cytosol, squalene and zeaxanthin freely migrated into the LD (in the -z direction). **C** Relative location of the migrating lipids with respect to the phospholipid monolayer. The location of the phospholipid headgroup is set to z = 0, and the range of the monolayer is highlighted in yellow. Once both lipids touched the cytosol-to-phospholipid boundary (Interface I), they rapidly penetrated the phospholipid monolayer. However, at the phospholipid-TAG boundary (Interface II), flexible worm-like squalene migrated deeper inside the TAG matrix than the rigid rod-like zeaxanthin, which stalled at the phospholipid-TAG boundary during the 500-ns simulation. **D** End-to-end distance $R_{end-to-end}$ during lipid migration. The length of squalene fluctuated continuously, exhibiting a wide range of sizes and shapes, whereas that of zeaxanthin remained constant due to its structural rigidity. **E** Estimation of friction coefficient ζ in neat TAGs obtained via steered MD simulations. Among heavily packed TAGs, worm-like squalene was subjected to twofold less frictional force than rod-like zeaxanthin. **F, G** Free energy landscape of (**F**) squalene and (**G**) zeaxanthin migration into cytosolic LDs, as calculated with multiwalker well-tempered metadynamics simulations. In the inset figures, the representative structures are displayed as linked with their corresponding order parameters (migration depth z and conformation order parameter μ) in the free energy landscape.

(iv) *sei1*, *loa1*, and *erd1* for membrane transformation (Fig. 3A, Supplementary Fig. 2, and Supplementary Note 1). Detailed information on the ten chosen genes is shown in Table 1. Previously, increasing the TAG or FA production[6,26], decreasing TAG hydrolysis[4,27], or delaying LD budding[28] have been effective methods for enlarging LDs. Despite few studies in which the number of LDs was controlled, promoting membrane budding is clearly helpful for increasing the number of LDs in a cell. We successfully overexpressed or deleted each of the chosen genes in a WT yeast strain for comprehensive evaluation of their specific effects on LD engineering (Fig. 3B, Supplementary Fig. 3, Supplementary Table 1, and Supplementary Note 1).

Next, LD regulatory genes were strategically combined to create two different designer yeast cells, in which either the LDs were much larger in size or the number of LDs was greater than in WT cells (Fig. 3C and Supplementary Note 1). In particular, to confirm the positional lipid storage in phospholipid-enveloped LDs, our gene combinations were focused on increasing the total volume or the net surface area of intracellular LDs compared to that of WT LDs (Fig. 3C, left). For the designer cell featuring a large LD size (LD-size), we chose three genes (*tgl3*, *tgl4*, and *lro1*) whose regulatory effects led to the largest LD volume (Fig. 3B and Supplementary Table 1) to build LD-size strain (WT-ΔTGL3 ΔTGL4 LRO1); compared to the number of LDs in the WT strain (23 LDs per cell), the number of LDs in the LD-size strain decreased by ~30%, but the average diameter of the latter LDs increased by ~70% ($D_{avg}$ ~ 0.33 μm), resulting in an ~220% increase in the total LD volume per cell (0.293 μm³) (Fig. 3C, middle). For the other designer cell with a large number of LDs (LD-number), two other genes (*loa1* and *erd1*) were chosen to generate the yeast strain (WT-LOA1 ΔERD1), and the combination of Loa1 overexpression with Erd1 deletion was highly synergistic. The LD size did not differ between strains ($D_{avg}$ ~ 0.20 μm), but in the LD-number strain, the number of LDs was greater, 51.67 per cell, which was ~130% higher than that of the WT strain, increasing the net surface area of these LDs by ~120% (6.495 μm² per cell) (Fig. 3C, right). Because of our LD engineering, the total lipid and LD lipid contents of LD-size and LD-number cells slightly increased compared to those of WT cells (Fig. 3D, inset), but the LD lipid compositions of the two engineered cells were quite similar each other (Fig. 3D, Supplementary Data 1 and 2), thereby excluding the possibility of improved LD storage by different LD lipid compositions.

**Lipid-dependent LD storage in our designer yeast cells**. Finally, to demonstrate that the structural flexibility of lipids is a major determinant of their migration and storage in LDs in vivo, we examined the storage dependency of intracellularly produced squalene and zeaxanthin in our designer cells (Fig. 4 and Supplementary Data 3). As our yeast cells were engineered to produce lipophilic squalene and zeaxanthin, we successfully guided

microbially produced membrane-impermeable terpenes to be transported to cytosolic LDs within the cell (Fig. 4A). Briefly, for squalene production in LD-size and LD-number strains, we overexpressed Erg20 and truncated Hmg1 (tHmg1), as deficiency of Hmg1 is known to be a metabolic bottleneck in the squalene biosynthesis pathway[21], thereby creating LD-size/SQ and LD-number/SQ cells, respectively. For zeaxanthin production, five different genes, *crtE*, *crtB*, *crtI*, *crtY*, and *crtZ*, originating from *Pantoea agglomerans*, were overexpressed[22], generating two yeast cells, LD-size/ZEA and LD-number/ZEA cells, were also prepared for use in the in vivo assessment of lipid storage.

Intracellular storage of squalene dramatically increased as the size of the LDs increased (Fig. 4B, left). When we quantified the amount of intracellular squalene in a time-dependent manner, the titers of squalene in the LD-size/SQ cells were significantly increased over time than those in the LD-number/SQ and the control yeast (WT/SQ) cells engineered for squalene production. For instance, after 144 h of cultivation, the squalene titer of the LD-size/SQ cells was 13.93 mg/L, which was ~3100% larger than that of the WT/SQ cells (0.43 mg/L). Interestingly, the amount of squalene storage exponentially increased as the total LD volume increased (~220%); the storage-boosting mechanism is not clear, but as the bulky TAGs with three fatty acids were diluted by single-chain squalene, TAG-packaging LDs may become leaky, thus further promoting squalene migration and accommodating a tremendous amount of squalene into the relaxed LD[18,29].

Rigid zeaxanthin showed a completely different storage-dependence pattern in our engineered yeast cells than that shown by flexible squalene; LD-number/ZEA cells stored more zeaxanthin than LD-size/ZEA cells, suggesting that the net surface of the LDs is a key factor in the intracellular storage of zeaxanthin (Fig. 4B, right). Moreover, the titer of zeaxanthin in the LD-number/ZEA cells was larger than that in the LD-size/ZEA cells and WT cells capable of zeaxanthin production (WT/ZEA). After 144 h of cultivation, for instance, the zeaxanthin titer of LD-number/ZEA (2.85 mg/L) was ~32% and ~124% larger than that of the LD-size/ZEA and WT/ZEA cells (2.16 and 1.27 mg/L, respectively). On the basis of this observation, we expected the zeaxanthin storage to be LD volume-independent. However, as the net LD surface area of the LD-number/ZEA cells was ~20% and ~120% larger than that of the LD-size/ZEA and WT/ZEA cells, we presumed that the increase in stored zeaxanthin correlated with the increase in the surface area of each LD, although membrane-bound organelles and cell membranes presumably also served as zeaxanthin-stored reservoirs[30].

The lipid-dependent storage pattern in LDs was consistent even when lipid productivity was further improved. As an inhibitor of squalene epoxidase, terbinafine can be used to block the synthesis of squalene epoxide from squalene, thereby accumulating squalene[31]; when the flux of the squalene processing is reduced, the amount of downstream mevalonate products,

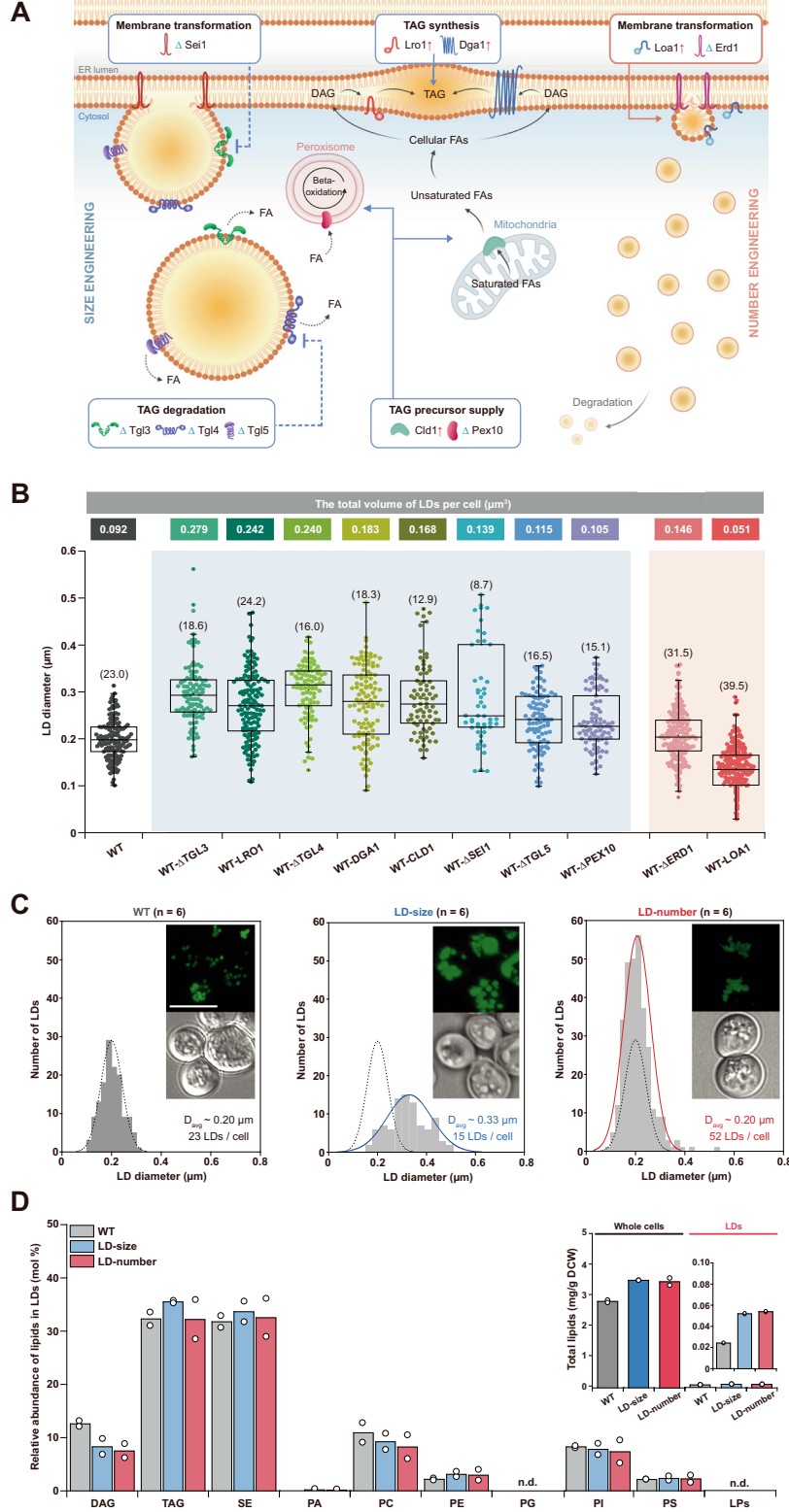

including zeaxanthin, can be also increased. In the presence of terbinafine (10 μg/mL), we investigated the storage levels of squalene and zeaxanthin, which were subsequently compared with those in the absence of terbinafine (Fig. 4B). Even though the terbinafine treatment was helpful to improve the overall lipid production for all different yeast cells, the distinctive storage patterns of squalene and zeaxanthin were not altered. Regardless

of terbinafine, the titers of flexible squalene for LD-size/SQ cells were much larger than those for LD-number/SQ cells, and rigid zeaxanthin showed larger storage in LD-number/ZEA cells than in LD-size/ZEA cells.

To further clarify the LD surface area effect in the storage of structurally rigid lipids, we prepared the yeast cells engineered to produce rigid β-carotene that is transported in intracellular LDs

**Fig. 3 Microbial LD engineering to control the size and number of cytosolic LDs. A** LD engineering strategy for controlling the size and number of LDs. By systematically rewiring LD biogenesis and degradation pathways, we controlled the size and number of LDs. We focused on genes associated with physicochemically distinct functions, including TAG synthesis, TAG degradation, TAG precursor supply, and membrane transformation, and their functions were intentionally promoted (solid line) or suppressed (dotted line). **B** Quantification of the size and number of LDs in yeast engineered by overexpression or deletion of each of ten select genes associated with LD engineering. The graph depicts scatter boxplots of morphological changes in LDs of individual gene-regulated yeast cells ($n = 6$). For the scatter plots, medians (horizontal line in box) with interquartile ranges (25–75%) are displayed. The numbers in parentheses indicate the LD number in each cell. **C** Frequency distribution histogram showing the size and number of LDs in the designer yeast cells ($n = 6$). From synergistic combinations of LD regulatory genes, we created two different designer yeast cells in which the LDs were much larger or in which there were more LDs (LD-size and LD-number, respectively) than the LDs of the WT cells. Confocal fluorescence microscopy images of the LDs in the designer yeast cells are shown as inset figures. Scale bar: 5 μm. **D** Comparative lipidomics of the designer yeast cells. Total lipid and LD lipid levels of LD-size and LD-number cells slightly increased compared to those of WT cells (an inset figure). When the LD-size and the LD-number cells were compared each other, their LD lipid compositions, along with the total lipid and the LD lipid levels, were quite similar each other. Relative enrichment of lipids in LDs is depicted in mol %. Yeast cells were grown in defined minimal media with 2% glucose at 30 °C for 24 h. n.d. not detectable. WT wild-type CEN.PK2-1D strain; DAG diacylglycerol; FA fatty acid; SE sterol ester; PA phosphatidate; PC phosphatidylcoline; PE phosphatidylethanolamine; PG phosphatidylglycerol; PI phosphatidylinositol; PS phosphatidylserine; LPs lyso-phospholipids. Source data are provided in the Source Data file.

**Table 1 Ten select genes associated with LD biogenesis and degradation.**

| Functions | Gene | Description | Engineering | Effect | Reference |
|---|---|---|---|---|---|
| TAG synthesis | lro1 | Acyltransferase that converts DAG to TAG | overexpression | Increased LD size | 47 |
| | dga1 | DAG acyltransferase | overexpression | Increased LD size | 48 |
| TAG degradation | tgl3 | Bifunctional TAG lipase and lysophosphatidylethanolamine acyltransferase | deletion | Increased LD size | 26 |
| | tgl4 | Multifunctional lipase/hydrolase/phospholipase | deletion | Increased LD size | 26 |
| | tgl5 | Bifunctional TAG lipase and LPA acyltransferase | deletion | Increased LD size | 26 |
| Precursor supply | cld1 | Mitochondrial cardiolipin-specific phospholipase | overexpression | Increased LD size | 47 |
| | pex10 | Peroxisomal membrane E3 ubiquitin ligase | deletion | Increased LD size | 49 |
| LD membrane transformation | sei1 | Seipin involved in LD assembly | deletion | Increased LD size | 50 |
| | loa1 | Lysophosphatidic acid acyltransferase | overexpression | Increased LD number | 51 |
| | erd1 | Predicted membrane protein required for lumenal ER protein retention | deletion | Increased LD number | 26 |

(Fig. 4C, Supplementary Fig. 4, and Supplementary Data 3). It is unclear whether a hydrophilic hydroxyl group at each end of the zeaxanthin chain (Fig. 1B, red) would affect the entrapment of the unfoldable lipids in the amphiphilic phospholipid layer of a LD; therefore, we additionally investigated the adsorption of fully hydrophobic β-carotene to confirm the rigidity-dependent lipid localization on the LD surface (Fig. 4C, orange). In line with our MD simulations and in vivo evaluation, β-carotene, a rod-like lipid, showed a greater affinity for the engineered cells with the maximal LD surface area (LD-number/BC cells) compared to the LD-enlarging cells (LD-size/BC cells) and the β-carotene-producing WT cells (WT/BC cells), regardless of terbinafine (Fig. 4C, left). Furthermore, using confocal microscopy, we directly visualized β-carotene localization on the surface of the intracellular LDs in yeast (Fig. 4C, right); β-carotene (red) was observed at the rim of multiple LDs (green), not in the core, proving that, in contrast to worm-like flexible lipids, rod-like rigid lipids cannot readily migrate into the LD core and reside at phospholipid membranes.

## Discussion

In this study, we questioned whether highly dynamic LD's lipid transportation is lipid-dependent, and we computationally and experimentally validated the finding that lipid chain flexibility determines a lipid migration ability into polymer melt-like TAGs. Our fundamental understanding of chain flexibility-dependent lipid migration indicates that living cell storage compartments are tailored to maximize target lipid storage within specific cells. Among medicinal terpenes, flexible squalene showed dramatically increased intracellular storage, by ~3100%, due to LD enlargement, while rigid zeaxanthin and β-carotene benefited from LD crowding. Compared to that of previous studies that simply focused on excess LD[18,19] (Supplementary Table 2), using a systematic LD engineering approach, we successfully customized the cells for improving the selective storage capacity of each lipid, and as observed by squalene migration into LDs, boosting lipid storage is a very interesting strategy to explore for more efficient metabolite reservoir remodeling.

Although our MD simulation simplified the complex process of lipid transportation across multiple boundaries and phases of intracellular LDs, it provided a guideline to determine the factors that can restrict the diffusion of target lipids in and out of LDs and characterize the architecture of compartmentalized LDs to engineer improved lipid storage and metabolism in desired organisms. We observed that the storage capability of rigid lipids (i.e., zeaxanthin and β-carotene) was not remarkably enhanced in our designer yeast cells, in which the net surface area of LDs had been increased; because the unfoldable lipids could not be stored inside these intracellular LDs, they were harbored by all kinds of membranes (e.g., organelles and plasma membranes). We note that protein crowding on the LD surface can interfere with lipid migration into LDs, not only by reduction of permeable surface areas, but also by alteration of LD monolayer properties (e.g., phospholipid packing, ordering, and organization)[3,32]. The limited penetration was expected because of the intertwined TAG chains, but this TAG density may be attenuated by reconstitution of the LD composition. For instance, increasing the proportion of fluidic lipids, such as cholesteryl esters, would facilitate lipid access into softened LDs and increase lipid penetration by lowering the free energy barrier to lipid diffusion.

Importantly, the application of our microbial LD engineering strategy is not limited to the storage of non-saponifiable lipids; it

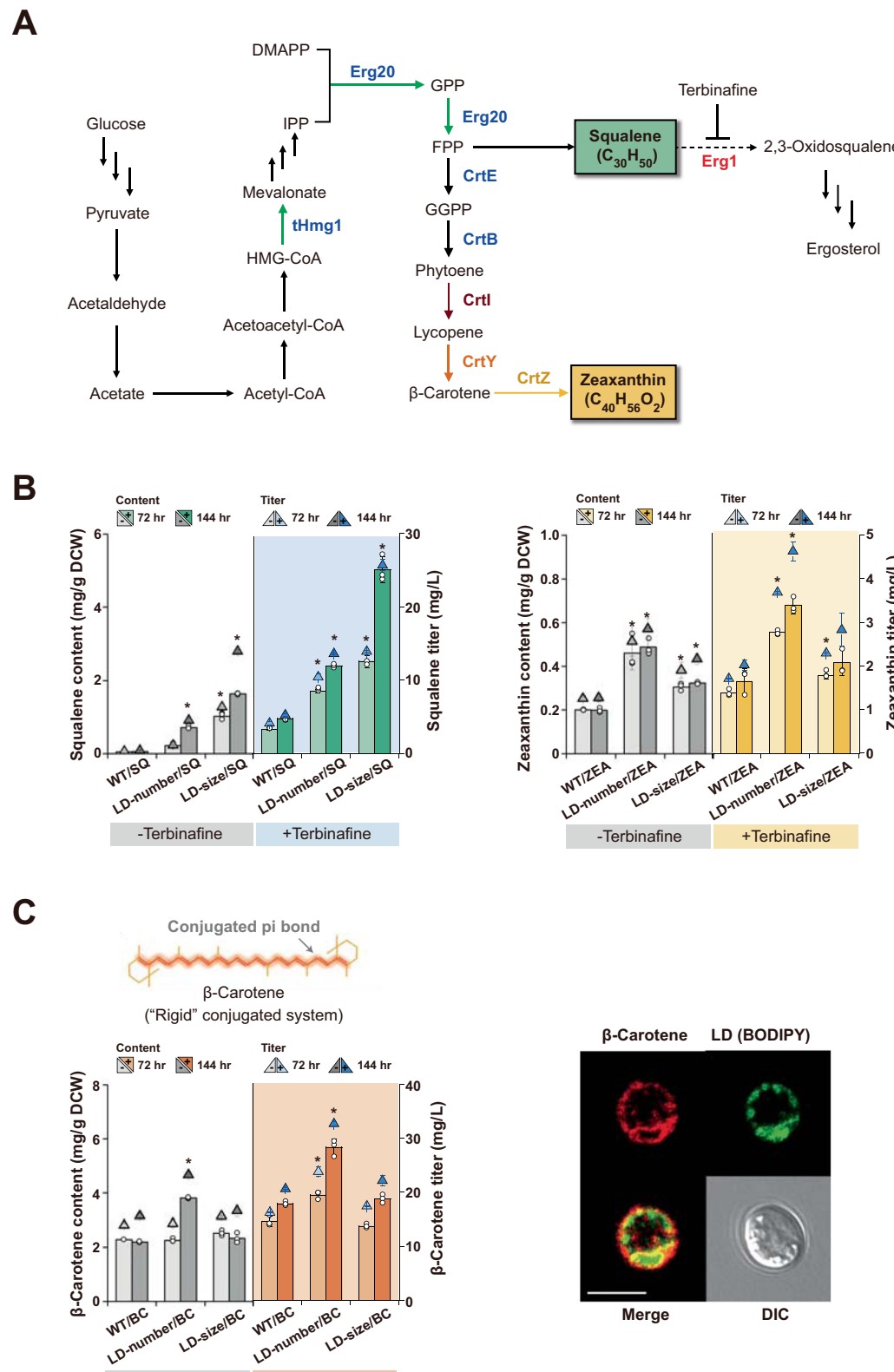

can be applied to other types of lipid transport systems and even metabolic systems. As the hydrophobicity and flexibility of lipids are easily predicted from their chemical structures, their compatible LDs can also be readily determined. Moreover, the surface localization of target lipids likely increases their opportunity to interact with many classes of LD surface proteins, mediating relevant metabolic processes. Moreover, as LDs represent well-

regulated oil-phase compartments within the aqueous cytosol of cells, selective partitioning of specific precursors, intermediates, and products of lipid metabolism may be achieved in these phase-separated organelles. Given the full control over the dynamics and functions of LDs, we envision that these unique phase-separated organelles will serve not only as warehouses with controlled entry and exit of various hydrophobic molecules,

**Fig. 4 In vivo evaluation of lipid-dependent migration and storage in LDs. A** The biosynthesis pathway of three non-saponifiable lipids, squalene, β-carotene, and zeaxanthin. Squalene was produced by overexpressing native squalene pathway genes (*thmg1* and *erg20*). β-carotene and zeaxanthin were produced via heterologous biosynthetic pathways upon expression of genes obtained from *Pantoea agglomerans* (*crtE*, *crtB*, *crtI*, *crtY*, and *crtZ*).
**B** Quantitative measurements of squalene and zeaxanthin produced by designer yeast cells. Structurally flexible squalene and rigid zeaxanthin showed a completely different storage-dependence pattern; the increase in squalene and zeaxanthin storage was significantly correlated with the increase in the total volume (size) and net surface area (number) of the LDs, respectively. **C** Localization of rigid β-carotene at the LD surface. Rod-like β-carotene showed a storage pattern in LDs very similar to that of zeaxanthin. Using confocal microscopy, we directly observed that β-carotene (red) was favorably adsorbed into all kinds of membranes, including the LD surface (green). Yeast cells were grown in defined minimal media with 2% glucose at 30 °C for 24 h. We note that β-carotene was inherently fluorescent (excitation at 450 nm and emission at 600 nm), and LDs were stained with the BODIPY fluorescent dye (excitation at 488 nm and emission at 500 nm). For production of target lipids, yeast cells were grown in defined minimal media with 2% glucose in the presence (+) or absence (−) of 10 μg/mL terbinafine at 30 °C for 144 h. The inhibition of squalene epoxidase (Erg1) by terbinafine was helpful to further increase production of squalene, zeaxanthin, and β-carotene, but it did not alter the distinctive storage patterns of the flexible and rigid lipids. All data represent the mean of biological triplicates, and error bars indicate the standard deviation. An asterisk (*) indicates that the value is significantly different ($P < 0.05$) from that for the respective control cells. Significance ($P$-value) was evaluated by two-sided $t$-test. IPP isopentenyl pyrophosphate; DMAPP dimethylallyl diphosphate; GPP geranyl pyrophosphate; FPP farnesyl pyrophosphate; GGPP geranylgeranyl pyrophosphate. Scale bar, 5 μm. Source data are provided in the Source Data file.

including nutraceutical products and therapeutic drugs[15], but also as hubs that promote the activation of metabolic pathways[16,29], rendering these LDs applicable for metabolic engineering and synthetic biology.

## Methods

**Molecular dynamics**. All MD simulations were performed using the massively parallel simulation package OpenMM version 7.5[33]. The phospholipids, TAGs and ions were modeled with the CHARMM36 force field[34] and a modified TIP3P water model[35], and compatible force field parameters for squalene and zeaxanthin molecules were determined using CgenFF[36]. The long-range electrostatics were calculated with the particle mesh Ewald (PME) method[37,38], and a cutoff distance of 1.2 nm was used for van der Waals interactions.

Initially, a solvated phospholipid bilayer composed of 60:40 1-palmitoyl-2-linoleoyl-sn-glycero-3-phosphatidylcholine:1,2-dipalmitoleoyl-sn-glycero-3-phosphoethanolamine dissolved in a 0.15 M KCl solution was prepared using CHARMM-GUI membrane builder[39] and equilibrated in isothermal-isobaric (NPT) ensemble. Subsequently, a separately equilibrated 10-nm-thick TAG slab was inserted between the upper and lower phospholipid monolayers: The snapshots presented in Fig. 2A, B show only the upper half of the simulation system. The combined system was composed of 256 phospholipids, 555 TAGs, ~10,500 water molecules and 54 ion pairs. After 10 ns of equilibration in the NPT ensemble at 300 K and 1 atm, the resulting system size was ~9 × 9 × 22 nm. The temperature was controlled with a Langevin thermostat, and the pressure was controlled with a Monte Carlo membrane barostat.

For unrestrained MD, a single squalene or zeaxanthin molecule was placed in the cytosolic water layer in the equilibrated slab system. A single, continuous long simulation trajectory of at least 500 ns was obtained for each of the squalene or zeaxanthin system with a canonical (NVT) ensemble at 300 K. The migration depth was defined as the $z$-axis location at the center of the mass of the lipophilic molecules with respect to the average position of the phospholipid head groups.

To obtain the free energy landscape upon the lipid migration through the two interfaces (Fig. 2F, G), multiwalker well-tempered metadynamics simulation with six replica walkers was performed for each of squalene and zeaxanthin system. Six initial representative structures were selected from unrestrained MD simulation trajectories with varying migration depths, in the range of $-2 \leq z \leq 1.5$; thus, each of the six simulations was initiated with a lipophilic molecule located in either the cytosol, interface I, or the phospholipid layer but not in the TAG layer. We employed two collective variables to characterize the migration depth ($z$) of the lipids and the conformation order parameter ($\mu$), defined as $z_{end-to-end}/R_{end-to-end}$, where $R_{end-to-end}$ is the end-to-end distance of the lipid molecule and $z_{end-to-end}$ is the z-axis component of the $R_{end-to-end}$ vector; $\mu$ represents a purely orientation transition of rigid zeaxanthin, and it represents changes in both the size and orientation of flexible squalene, as illustrated in the inset of Fig. 2F, G. During metadynamics simulations, the biasing potential changes adaptive to the instantaneous value of the collective variables, allowing flexible transitions requiring simultaneous changes of multiple collective variables. The selective lipid migration pathway drawn as arrows in Fig. 2F highlights this strength of metadynamics simulation.

The friction caused by multi-tail TAG layer was compared by performing steered MD simulations for a single squalene or zeaxanthin molecule placed in a neat TAG layer composed of 745 TAG molecules (Fig. 2E). After 10 ns equilibration at 300 K and 1 atm, a constant pulling force was applied to the lipid molecule and was increased stepwise in 20-ns intervals until the lipid molecule showed drift motion along the pulling force with constant velocity, which occurred at a force constant of 200 kJ/mol/nm. The friction coefficient of the lipid molecule was calculated using the Langevin equation $\zeta = -F/mv$, where $F$ is the pulling force and $m$ and $v$ are the mass and the drift velocity of the lipid, respectively.

**Plasmids and strain construction**. All plasmids, strains and primers used in this study are listed in Supplementary Data 4 and 5. The plasmids used in this study were generated via insertion of a gene fragment, which had been obtained either from yeast genomic DNA or stock plasmids and amplified with corresponding primer pairs and digested with restriction enzymes, into pUC57-URA3-derived vectors for strain construction or p416-derived vectors for gene overexpression. Gene deletion and modification were introduced via the URA3-blaster genetic disruption method[40]. Recombination cassettes for deleting and integrating a single gene were amplified by PCR from pUC57-URA3- and pUC57-URA3-derived vectors containing the gene of interest, respectively, with primer pairs introducing regions homologous to the target recombination site. The standard LiAc/ssDNA/PEG method was used for yeast transformation[41].

**Confocal fluorescence microscopy**. Yeast cells were cultivated in yeast synthetic complete (YSC) medium (0.19% yeast synthetic dropout medium without uracil and 0.67% yeast nitrogen base without amino acids) supplemented with 2% (w/v) glucose. After growing for 24 h at 30 °C with 250 rpm shaking, the cells were fixed for 20 min with 3.7% formaldehyde in phosphate-buffered saline (pH 7.4) and stained with 5 μM BODIPY 493/503 dye for 30 min at 30 °C. Then, the cells were washed twice with 1x phosphate-buffered saline and observed with a Zeiss-LSM 780 multiphoton confocal microscope (Zeiss, Germany) equipped with a Plan-Apochromat 63x/1.4 NA oil immersion objective. Confocal images were analyzed using ImageJ software 1.8 (National Institutes of Health, USA) and ZEN imaging software 2.1 (Zeiss, Germany).

**Lipidomics analysis**. Lipidomics analysis was performed as described previously[42,43]. In brief, yeast cells were cultivated in YSC medium supplemented with 2% (w/v) glucose. After growing for 24 h at 30 °C with 250 rpm shaking, 20 OD equivalents of the cells were harvested by centrifugation ($3500 \times g$, 5 min, 4 °C). Mass spectrometry-based lipid analysis was performed by Lipotype GmbH (Dresden, Germany). Lipids were extracted using chloroform/methanol procedure[42]. Samples were spiked with lipid class-specific internal standards. Lipid extracts were analyzed on a hybrid quadrupole/Orbitrap mass spectrometer equipped with an automated nano-flow electrospray ion source in both positive and negative ion mode.

Total lipid weight was determined following a previously reported method[44]. Specifically, 10 OD equivalents of the cells were harvested by centrifugation ($3500 \times g$, 5 min, 4 °C). The harvested cells were resuspended in 6 mL of chloroform/methanol (1:1 volumetric) and crushed using a FastPrep-24 5 G homogenizer (MP Biomedicals, USA). The samples were then mixed with 1.5 mL water and vortexed for 1 min. After centrifugation, the organic layer was collected, washed with 1.5 mL of 0.1% (w/v) NaCl water solution, and dried at room temperature overnight. The tube was further dried in an oven at 80 °C and then weighed to determine total lipid weight.

**Terpene production by flask fermentation**. Yeast cells were grown in 10 mL of YSC medium supplemented with 2% (w/v) glucose for the production of terpenes (squalene, zeaxanthin and β-carotene). After overnight cultivation at 30 °C with 250 rpm shaking, the seed cultures were inoculated into 250-mL flasks containing 50 mL of YSC medium supplemented with 2% (w/v) glucose in the presence or absence of terbinafine (to an initial OD$_{600}$ of 0.5). Flask fermentation was carried out at 30 °C and 250 rpm for 6 days. All flask fermentation procedures were repeated in three independent experiments.

**Terpene extraction and quantification**. Squalene, zeaxanthin, and β-carotene titers were quantified as described previously[21,45]. Briefly, yeast cells were harvested by centrifugation at $13,000 \times g$ for 5 min. The harvested cells were resuspended in 0.6 mL of a 1:1 methanol-acetone (MA) solution with lysing matrix C (MP Biomedicals, USA). The mixture was mechanically disrupted using a FastPrep-24 5 G homogenizer (MP Biomedicals, USA) according to the manufacturer's instructions. After filtration using 0.2-μm syringe filters, the terpenes extracted from the MA solution were analyzed using an Agilent HPLC system equipped with a UV detector (squalene was detected at 203 nm, and zeaxanthin and β-carotene were detected at 450 nm). The samples were separated on a Kinetex 5 μm EVO C18 column (Phenomenex, Aschaffenburg, Germany) at 30 °C using isocratic elution at a flow rate of 1.0 mL/min for 30 min.

**Figure preparation**. Figures were prepared using BioRender.Com for scientific illustrations. MD simulation snapshots in Fig. 2 and Supplementary Movie 1 and 2 showing the simulation trajectories were prepared using visual molecular dynamics software version 1.9.4[46].

**Statistics and reproducibility**. The confocal fluorescence microscopy experiments in Figs. 3C; 4C and in Supplementary Figs. 3; 4 were performed at least triplicate at two independent times.

**Reporting summary**. Further information on research design is available in the Nature Research Reporting Summary linked to this article.

## Data availability

Data supporting the findings of this work are available within the paper and its Supplementary Information files. A reporting summary for this article is available as a Supplementary Information file. Source data are provided with this paper.

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

## Acknowledgements

This work was supported by the "Development of next-generation biorefinery platform technologies for leading bio-based chemicals industry" project (NRF-2022M3J5A1056072), the "Development of an integrated process to produce lig-nocellulosic biomass–derived fermentable sugars for next-generation biorefinery" project (NRF-2022M3J5A1056173), the Bio & Medical Technology Development Program (NRF-2022M3A9B6082671), and the Basic Science Research Program (NRF-2021R1A4A1032162 and NRF-2021R1C1C1009323) through the National Research Foundation of Korea (NRF) grant funded by the Ministry of Science, ICT (MSIT). S.-H.S. and J.Y.L. acknowledge funding from Korea Research Institute of Chemical Technology through Core Program (SS2242-10).

## Author contributions

S.-H.S., C.Y.S., S.S.O., and J.Y.L. conceived the study and designed the experiments. G.P. and J.L. performed the molecular dynamics simulation. All authors assisted in performing the experiments and data analysis. J.Y.L. supervised the research. S.-H.S., C.Y.S., S.S.O., and J.Y.L. wrote the manuscript.

## Competing interests

The authors declare no competing interests.
