## [Peer Review File · Nature Communications]

Chain flexibility of medicinal lipids determines their selective partitioning into lipid dropletsReviewers' Comments:

Reviewer #1:

Remarks to the Author:

all comments are in the attached Word document

A strength of this work is the combination of experimental and computational techniques to examine the molecular interactions and dynamics between free fatty acids and lipid droplets. The study aims to determine the most important factors that modulate internalization of specific molecules into lipid droplets, and emphasizes the need and interest in understanding these topics for pharmaceutical and engineering applications. The authors use molecular dynamics, biased and unbiased, to characterize the interaction of a flexible lipophilic molecule and a rigid one. They are able to show the chemical nature of these compounds is key to determining their mode of interaction and uptake into lipid droplets. The simulations offer a detailed overview of the conformational changes in each lipophilic molecule as they interact with a lipid monolayer that represents the lipid droplet interface. With the insights gained from simulation, the study examined the process in-vivo to corroborate the relationship between chemical structure of the lipophilic molecule and uptake mode as well as the influence in size and content of the lipid droplet in the process. Experimental results follow the expected trends suggested by simulation.

The study is relevant and contributes new insights about the operation and regulation of lipid storage in lipid droplets. The interest in lipid droplets has increased in the past decade, and still few works focus on explicitly understanding the biophysical aspects that contribute to the use and regulation of these lipid bodies. I support the publication of the article with minor revisions as per my comments below.

Comments:

1. The article focuses on studies to determine key factors that modulate lipid uptake and degree of internalization into lipid droplets. The material presented is very timely and relevant, but the title of the manuscript is rather misleading.
2. In figure 3, and the corresponding discussion in the main paper, the authors claim their experimental protocol reproduces LDs of different size, which can be used to study the effect of storage space on lipid uptake. The data presented in Figure 3.B shows differences in the mean value; I'd like the authors to discuss some of the implications of the distributions of their data. It seems that some of the mutants are not really statistically different.
3. The authors acknowledge the simulated systems are a simplified version of the system and do validate the conclusions from the computational models with experimental data. LD regulation is certainly not trivial, and the article provides valuable insights.
 - a. How would the composition of the LD monolayer itself modulate lipid uptake? Noting this may change across organisms.
 - b. It is important to also mention in this discussion the role (or not?) of protein crowding in the LDs. How proteins on the monolayer may also interfere with uptake and interactions of free FA or other lipids in the cytosol and the LD.
4. In the methods section, were multiple replicas run for the unrestrained MD trajectories or only one trajectory of 500ns?
5. Keep the name consistent for the MD biasing method for reader ease. Well-tempered metadynamics is mentioned in the results, and then "steered MD" are detailed in the methods section.
 - a. Since the uptake of lipophilic molecules is observed on the unrestrained MD trajectories, could "plain" umbrella sampling be used instead using the sampled configurations as coordinates for the windows to compute the PMF for the process? What is the advantage of using well-tempered metadynamics?

Reviewer #2:

Remarks to the Author:

This study dissects the molecular determinants of hydrophobic molecule targeting to lipid droplets. Using molecular simulations and some yeast biology, they show that hydrophobic biomolecules with different molecular flexibilities will differentially integrate into the lipid droplet monolayer surface and isotropic neutral lipid core. Using some yeast mutants that effect the LD size and/or abundance in budding yeast cells, they also attempt to examine how changes to LD size/abundance influence hydrophobic accumulation of these molecules.

While the molecular dynamics simulations are informative, there are several major concerns with the yeast based work. A general concern is that many of the yeast mutations utilized have pleiotropic effects beyond simply LD size or abundance. Since these alterations are not considered in many of the experiments, it is hard to make conclusions about several of the key experiments (see details below). Numerous controls for these experiments are also lacking.

A second concern is the writing, which is vague and generally hard to follow. Important details in the yeast section on which mutant strain is being used in which experiment are essential to understanding the experiments, yet are not stated.

A third concern is the lack of conceptual advance of the study. It is generally accepted that larger LDs will house more hydrophobic compounds. How the molecular shape of the hydrophobic molecule impacts LD delivery/retention is an important question, but the yeast based work here fails to fully address this at present. The lack of significant conceptual advance makes Nature Communications inappropriate for this manuscript.

Major concerns:

1) Some of the mutant yeast strains make interpreting experiments challenging as they can have multiple effects when these genes they are deleted. Particular examples are mutants like ERD1. ERD1 is a poorly defined ER protein, and its deletion causes ER protein retention to be altered (Hardwich, EMBO J, 1990). The effect ERD1 loss has on LD size/abundance is likely indirect. Over-expressing LOA1 in this background likely causes even more changes in ER and LD homeostasis, and this needs to be addressed prior to interpretation of the experimental data.

Other mutations that are used here may cause multiple changes to LD lipid composition itself. Over-expression of LRO1 in the *tgl3,4*-double KO background will increase LD size, but LRO1 itself is a phospholipid:DAG acyltransferase. Overexpressing it is thus also increasing the pool of lyso-phospholipids as well as increasing TAG pools, and both these changes could drastically alter LD lipid composition and the properties of hydrophobic molecules that target to LDs.

2) In Figure 4, squalene accumulation is monitored over time in various conditions and shown to increase with LD size. Zeaxanthin did not accumulate as squalene did. It is not clear whether the yeast in these experiments are converting squalene to squalene epoxide or downstream mevalonate products at the same rate. This is a key control, and needs to be investigated. Squalene processing can also be inhibited by terbinafine, and may serve as a good control for this experiment.

Reviewer #3:

Remarks to the Author:

The manuscript by Son et al. reports that the storage capacity/accumulation of lipophilic products by engineered yeast can be affected by the structural properties of the lipophilic products and the size/surface area of the lipid droplets in the yeast cells. The authors used MD simulations to demonstrate the differential deposition locations of squalene and beta-carotene in lipid droplets. The

worm-like flexible structure of squalene enabled deeper penetration into the core of lipid droplets. In contrast, the rod-like rigid structure of zeaxanthin and beta-carotene led to the trapping of the zeaxanthin and beta-carotene on the surface of lipid droplets. To confirm the differential localization of squalene, beta-carotene, and zeaxanthin in vivo, the authors constructed *Saccharomyces cerevisiae* mutant strains containing varied numbers and sizes of lipid droplets. Two mutant *S. cerevisiae* strains with increased sizes (LD-size) and numbers (LD-number) of lipid droplets were selected and used to examine squalene, beta-carotene, and zeaxanthin production. As predicted by the MD simulations, the LD-size strain was superior to the LD-number strain for squalene accumulation. On the contrary, the LD-number strain was slightly better than the LD-size strain for accumulating beta-carotene and squalene.

The storage capacity (mainly in terms of lipid content) of hydrophobic molecules in engineered yeast has been pointed as a potential limiting factor for industrial production. However, the size and number of lipid droplets have not been considered for the enhanced production of hydrophobic molecules. Therefore, the main idea of the study is new and interesting but in vivo validation will need more rigorous experiments to support the main argument of the study.

1. The authors did not measure the total lipid contents of their engineered yeast strains. While the number and size of lipid droplets are the main targets of their engineering, total lipid content has been reported as a primary differentiator for the enhanced storage of hydrophobic molecules. As such, the authors will need to show the total lipid contents of LD-size and LD-number strains are equivalent. Then the higher accumulation of squalene in the LD-size than the LD-number strain can be attributed to the structural properties of squalene and LDs. This reviewer suggests including the lipid contents of the engineered strains shown in Figure B.
2. P-values from statistical testing of the product concentrations in Figures 4B and 4C need to be presented.
3. The presented squalene concentrations by WT/SQ, LD-size/SQ, and LD-number/SQ in Figure 4B were much lower than the reported concentrations (30-150 mg/L) in other studies. As such, the statement in the abstract, "we observed that intracellular storage of squalene was dramatically increased, by ~3100%, with LD volume expansion" might be exaggerating. Also, the Production comparison in the Supplementary Table 3 might not be correct.
4. The authors are using only titers of squalene, beta-carotene, and zeaxanthin (mg/L) for comparing the engineered strains. As the lipophilic products are intracellular products, the specific contents (mg/g cell) need to be also measured and considered for the comparison as in the previous studies.
5. The authors will need to describe when the cells were harvested for obtaining microscopic images in Figure 4C and how beta-carotene was visualized.

Point-by-Point Response

Revised Title: Chain flexibility of medicinal lipids determines their selective partitioning into lipid droplets

Corresponding Authors: Ju Young Lee & Seung Soo Oh & Chang Yun Son

Authors: So-Hee Son et al.

Reviewer #1

Reviewer overall comment:

A strength of this work is the combination of experimental and computational techniques to examine the molecular interactions and dynamics between free fatty acids and lipid droplets. The study aims to determine the most important factors that modulate internalization of specific molecules into lipid droplets, and emphasizes the need and interest in understanding these topics for pharmaceutical and engineering applications. The authors use molecular dynamics, biased and unbiased, to characterize the interaction of a flexible lipophilic molecule and a rigid one. They are able to show the chemical nature of these compounds is key to determining their mode of interaction and uptake into lipid droplets. The simulations offer a detailed overview of the conformational changes in each lipophilic molecule as they interact with a lipid monolayer that represents the lipid droplet interface. With the insights gained from simulation, the study examined the process in-vivo to corroborate the relationship between chemical structure of the lipophilic molecule and uptake mode as well as the influence in size and content of the lipid droplet in the process. Experimental results follow the expected trends suggested by simulation. The study is relevant and contributes new insights about the operation and regulation of lipid storage in lipid droplets. The interest in lipid droplets has increased in the past decade, and still few works focus on explicitly understanding the biophysical aspects that contribute to the use and regulation of these lipid bodies. I support the publication of the article with minor revisions as per my comments below.

Overall response: We sincerely thank you for understanding the significance of our work and are grateful for your encouraging comments and exceptionally useful suggestions that helped us to further strengthen the manuscript. We tried our best to answer all your questions in the point-by-point responses as listed below. The changes in the revised manuscript were

highlighted in red accordingly.

Reviewer comment 1:

The article focuses on studies to determine key factors that modulate lipid uptake and degree of internalization into lipid droplets. The material presented is very timely and relevant, but the title of the manuscript is rather misleading.

Response: We greatly appreciate the reviewer's important suggestion. Relying on the chain flexibility of lipids, their selective partitioning into lipid droplets could be determined, which is exactly the material that this manuscript presented about. Instead of the vague and misleading title, we newly introduced the title with the clear meaning, "**chain flexibility of medicinal lipids determines their selective partitioning into lipid droplets.**"

Reviewer comment 2:

In figure 3, and the corresponding discussion in the main paper, the authors claim their experimental protocol reproduces LDs of different size, which can be used to study the effect of storage space on lipid uptake. The data presented in Figure 3B shows differences in the mean value; I'd like the authors to discuss some of the implications of the distributions of their data. It seems that some of the mutants are not really statistically different.

Response: We thank the reviewer for the valuable comment that can strengthen our manuscript. In increasing the size of LDs, we individually tested eight different genes that have been confirmed to influence on the LD size by previous studies (Table 1). The physicochemical functions of the genes are distinct, and the resulting LD biogenesis and degradation are affected in different extents, causing the LD size distribution to be varied for the different gene modification. Statistically, the individual gene modification displayed the LD size difference compared to no modification; when each of the eight genes were individually deleted or overexpressed in the wild-type (WT) yeast strain, all selected genes yielded modest (1.2 to 1.6-fold) increase in the LD size compared with the WT strain (**Figure 3B** and **Supplementary Table 1**). We further clarified the individual gene effect in increasing the size of LDs in new **Supplementary Note 1** in the revised Supplementary Information.

Reviewer comment 3:

The authors acknowledge the simulated systems are a simplified version of the system and do validate the conclusions from the computational models with experimental data. LD regulation is certainly not trivial, and the article provides valuable insights.

- a. How would the composition of the LD monolayer itself modulate lipid uptake? Noting this may change across organisms.
- b. It is important to also mention in this discussion the role (or not?) of protein crowding in the LDs. How proteins on the monolayer may also interfere with uptake and interactions of free FA or other lipids in the cytosol and the LD.

Response: We gratefully thank the reviewer's insightful comments. With regard to the composition of the LD monolayer that the reviewer pointed out in Comment a, it may modulate the packing and curvature of the LD monolayer, affecting lipid uptake of LDs. However, the LD compositions are not significantly different among almost all organisms from bacteria to humans [1, 2]; in general, the LD monolayer is mainly enriched in phosphatidylcholine, phosphatidylethanolamine or phosphatidylinositol [3]. By MD simulations, it was demonstrated that the lipid uptake into the LD monolayer would be mainly due to hydrophobic interactions in yeast, so we believe that the similar lipid uptake would be observed across almost all organisms.

As the reviewer pointed out in Comment b, we strongly agree that the LD surface proteins would influence on the degree of lipid uptake by LDs. On the surface of LDs, crowded proteins can disturb lipids to diffuse into the core of LDs by reducing hydrophobic interaction areas, and the protein crowding on the LD surface is also known to locally disturb LD monolayer properties, such as lipid packing, lipid ordering and lipid organization, all of which can affect the diffusion of lipids into LDs [4, 5]. As suggested by the reviewer, we included the discussion in the revised manuscript.

Page 16, Line 323: ... harbored by all kinds of membranes (*e.g.*, organelles and plasma membranes). **We note that protein crowding on the LD surface can interfere with lipid migration into LDs, not only by reduction of permeable surface areas, but also by alteration of LD monolayer properties (*e.g.*, phospholipid packing, ordering, and organization)^{3,32}.**

[1] Zhang C *et al.* The lipid droplet: A conserved cellular organelle. 2017. *Protein Cell.* 8, 796-800

[2] Zhe Cao *et al.* Dietary fatty acids promote lipid droplet diversity through seipin enrichment in an ER subdomain. 2019. *Nat Commun.* 10:2902

[3] Zhe Cao *et al.* Dietary fatty acids promote lipid droplet diversity through seipin enrichment in an

[4] René Bartz *et al.* Lipidomics reveals that adiposomes store ether lipids and mediate phospholipid traffic. 2007. *J Lipid Res.* 4, 837-847

[5] Lucie Caillon *et al.* Triacylglycerols sequester monotopic membrane proteins to lipid droplets. 2020. *Nat Commun.* 11, 3944

Reviewer comment 4:

In the methods section, were multiple replicas run for the unrestrained MD trajectories or only one trajectory of 500ns?

Response: A single, continuous long trajectory was obtained with unrestrained MD simulations for each of the squalene and zeaxanthin. The multi-walker well-tempered metadynamics simulations implemented six replicas running in parallel for each of the squalene and zeaxanthin. To clarify this, we modified the method section as follows.

Page 18, Line 367: For unrestrained MD, a single squalene or zeaxanthin molecule was placed in the cytosolic water layer in the equilibrated slab system. **A single, continuous long simulation trajectory of at least 500 ns was obtained for each of the squalene or zeaxanthin system with a canonical (NVT) ensemble at 300 K.**

Reviewer comment 5:

Keep the name consistent for the MD biasing method for reader ease. Well-tempered metadynamics is mentioned in the results, and then “steered MD” are detailed in the methods section.

- a. Since the uptake of lipophilic molecules is observed on the unrestrained MD trajectories, could “plain” umbrella sampling be used instead using the sampled configurations as coordinates for the windows to compute the PMF for the process? What is the advantage of using well-tempered metadynamics?

Response: We are sorry for our unclear notion of MD biasing method. Both well-tempered

metadynamics and the steered MD simulations were performed to obtain the 2D free energy map in the slab system and the friction coefficient in neat TAG, respectively. We now clarify this in the method section:

Page 9, Line 153: To identify this lipid-dependent diffusion barrier, we measured the solute-dependent friction coefficient in equilibrated neat TAGs by steered MD simulations²⁴ (**Figure 2E**).

Page 19, Line 373: **To obtain the free energy landscape upon the lipid migration through the two interfaces (Figure 2F and G), multiwalker well-tempered metadynamics simulation with six replica walkers was performed for each of squalene and zeaxanthin system.** Six initial representative structures were selected from unrestrained MD simulation trajectories with varying migration depths, in the range of $-2 \leq z \leq 1.5$;

Page 19, Line 389: **The friction caused by multi-tail TAG layer was compared by performing steered MD simulations for a single squalene or zeaxanthin molecule placed in a neat TAG layer composed of 745 TAG molecules (Figure 2E).** After 10 ns equilibration at 300 K and 1 atm, ...

Reviewer comment 5a: Since the uptake of lipophilic molecules is observed on the unrestrained MD trajectories, could “plain” umbrella sampling be used instead using the sampled configurations as coordinates for the windows to compute the PMF for the process? What is the advantage of using well-tempered metadynamics?

Response: In the unrestrained MD simulations, spontaneous uptake of lipophilic molecules was observed for both squalene and zeaxanthin at the first cytosol-phospholipid interface, but full penetration into the TAG core layer was observed only for the flexible squalene molecule. Moreover, the free energy landscape across the interface was strongly dependent on the orientation and the conformation of the lipid molecules as illustrated in **Figure 2F and G**. Thus, it was essential to capture the free energy differences associated with both the migration depth as well as the changes in orientation and conformation of the lipid molecules. We initially tried plain umbrella sampling based on the migration depth. Restricting the position of lipid molecules in narrow window was required for converged free energy estimate in such simulations. But this caused the conformation and orientation of the lipid molecules trapped in the local free energy minimum and did not sample proper transition in higher dimensional space

which required the simultaneous changes in both migration depth and conformation order. On the other hand, using well-tempered metadynamics simulations allowed free transition in the complex high-dimensional free energy landscape by interactive changes of biasing potential, which revealed the high-dimensional transition path observed for flexible squalene molecule as illustrated in **Figure 2G**. This highlights the strength of free energy methods using adaptive biasing potential. We modified the method section to emphasize on this.

Page 19, Line 383: ... and it represents changes in both the size and orientation of flexible squalene, as illustrated in the inset of **Figure 2F** and **G**. **During metadynamics simulations, the biasing potential changes adaptive to the instantaneous value of the collective variables, allowing flexible transitions requiring simultaneous changes of multiple collective variables. The selective lipid migration pathway drawn as arrows in **Figure 2F** highlights this strength of metadynamics simulation.**

Reviewer #2

Reviewer overall comment:

This study dissects the molecular determinants of hydrophobic molecule targeting to lipid droplets. Using molecular simulations and some yeast biology, they show that hydrophobic biomolecules with different molecular flexibilities will differentially integrate into the lipid droplet monolayer surface and isotropic neutral lipid core. Using some yeast mutants that effect the LD size and/or abundance in budding yeast cells, they also attempt to examine how changes to LD size/abundance influence hydrophobic accumulation of these molecules.

While the molecular dynamics simulations are informative, there are several major concerns with the yeast based work.

Overall response: We greatly appreciate your invaluable comments on our work that explored fundamental understanding of chain flexibility-dependent lipid migration into lipid droplets. Your concerns on our yeast engineering were significantly important for us to strengthen our manuscript, and we tried our best to carefully, diligently, and patiently address all your insightful comments and suggestions in a point-by-point manner. As below, we thoroughly prepared the point-by-point response and the revised manuscript and Supplementary Information, and the changes in the revised manuscript were highlighted in red accordingly. As we answered all your helpful questions, we strongly believe that the clarity and impact of our paper is further improved.

General concern 1:

A general concern is that many of the yeast mutations utilized have pleiotropic effects beyond simply LD size or abundance. Since these alterations are not considered in many of the experiments, it is hard to make conclusions about several of the key experiments (see details below). Numerous controls for these experiments are also lacking.

Response: We understand well the reviewer's concern about pleiotropic mutations and apologize that detailed explanation and rationale of our yeast engineering were not included in the original version of our manuscript because the main text is limited to 5,000 words only. We carefully identified ten different genes, of which effects on the LD number or size have been validated by previous studies (**Table 1**), and we reconfirmed each gene's regulatory effect one by one (**Figure 3B**, **Supplementary Figure 3**, and **Supplementary Table 1**). Moreover, to

create two different designer yeast cells, we combined only two or three of the most effective genes and subsequently validated that the LD size and number engineering led to the largest LD volume for confirmation of flexible lipid storage into LDs and the widest LD surface area for observation of rigid lipid retention onto the LDs, respectively. We have now further clarified the detailed explanation and rationale of our yeast engineering in new **Supplementary Note 1** in the revised Supplementary Information. Furthermore, as the reviewer suggested, to ensure the effectiveness of our LD engineering in supporting our claim, lipid-dependent positional storage in LDs, we have conducted additional analyses, including 1) mass spectrometry-based lipidomic profiling to confirm no change of LD compositions between the engineered cells and 2) terpene production improvement by use of terbinafine to observe the productivity-irrelevant trend of lipid-dependent storage in LDs.

General concern 2:

A second concern is the writing, which is vague and generally hard to follow. Important details in the yeast section on which mutant strain is being used in which experiment are essential to understanding the experiments, yet are not stated.

Response: We sincerely apologize for the lack of clarity. Once again, because the main text is limited to 5,000 words only, the yeast-relevant experiments were briefly described only in the original version of manuscript. We strongly agree with the reviewer that the important details of our yeast engineering should be included for better understanding; the accurate and detailed information about our LD size and number engineering was newly included as **Supplementary Note 1** in the revised Supplementary Information. We believe that this note would be helpful to understand how we rationally selected ten different genes for LD size and number engineering, and how our yeast mutant strains (LD-size and LD-number cells) were successively constructed from the wild-type (WT) strain.

General concern 3:

A third concern is the lack of conceptual advance of the study. It is generally accepted that larger LDs will house more hydrophobic compounds. How the molecular shape of the hydrophobic molecule impacts LD delivery/retention is an important question, but the yeast based work here fails to fully address this at present. The lack of significant conceptual advance

makes Nature Communications inappropriate for this manuscript.

Response: We agree with the reviewer that “larger LDs will house more hydrophobic compounds” is a generally accepted concept; for instance, by promoting synthesis of lipids, such as TAGs, a previous study has reported that LDs could be enlarged to aid the accumulation of lipophilic natural product lycopene in *S. cerevisiae* [1]. As the reviewer expected, the molecular shape would be important to determine migration of lipids into LDs or their retention on the LD surface, but unexpectedly, our work showed that the selective partitioning of the hydrophobic lipids into the hydrophobic LDs could be driven by NEITHER nonspecific hydrophobic interactions NOR specific molecular recognition, all of which are relevant to the molecular shape of lipids. By combining molecular dynamics (MD) simulations and yeast LD engineering, we demonstrated for the first time that the structural flexibility of lipids is one of decisive factors in selective diffusion and consequent partitioning of lipids into LDs. As mentioned by the Reviewer #3, “the main idea of the study is new and interesting”, and the fundamental understanding of the LD’s lipid storage would provide “new insights about the operation and regulation of lipid storage in lipid droplets” as mentioned by the Reviewer #1.

This proof-of-concept study tried to validate the hypothesis, “lipid migration into lipid droplets can depend on chain flexibility of the lipids,” as claimed by our MD simulations, but we agree with the reviewer about the necessity of additional yeast-based experiments. Therefore, first, we performed the mass spectrometry-based lipid analysis and confirmed that the lipid compositions of LDs were not altered even after LD gene modifications, supporting that the lipid uptake was mainly attributed to the inherent property of the lipid, not the altered property of the LD. Second, we also examined the effect of improved lipid production using terbinafine, an inhibitor of squalene epoxidase, and observed that even with higher productions of squalene, zeaxanthin, and β -carotene, the trends of lipid-dependent LD storage did not change, suggesting that the chain flexibility of lipids is indeed relevant to their selective partitioning into LDs. Once again, using confocal microscopy, we directly visualized rigid β -carotene localization on the LD surface, whereas flexible squalene was found inside the LD. Along with the additional proofs and experiments, we believe that the significance and the quality of our work would be further improved for its publication.

[1] Ma *et al.* Lipid engineering combined with systematic metabolic engineering of *Saccharomyces cerevisiae* for high-yield production of lycopene. 2019. *Metab Eng.* 52, 134–142

-Major Concern-

Major concern 1:

Some of the mutant yeast strains make interpreting experiments challenging as they can have multiple effects when these genes they are deleted. Particular examples are mutants like ERD1. ERD1 is a poorly defined ER protein, and its deletion causes ER protein retention to be altered (Hardwich, EMBO J, 1990). The effect ERD1 loss has on LD size/abundance is likely indirect. Over-expressing LOA1 in this background likely causes even more changes in ER and LD homeostasis, and this needs to be addressed prior to interpretation of the experimental data.

Other mutations that are used here may cause multiple changes to LD lipid composition itself. Over-expression of LRO1 in the *tgl3,4*-double KO background will increase LD size, but LRO1 itself is a phospholipid:DAG acyltransferase. Overexpressing it is thus also increasing the pool of lyso-phospholipids as well as increasing TAG pools, and both these changes could drastically alter LD lipid composition and the properties of hydrophobic molecules that target to LDs.

Response: We thank the reviewer for the valuable inquiries. We agree that some genes used in this study can have pleiotropic functions; in addition to LD biogenesis and degradation, the mutation (*i.e.*, overexpression or deletion) of the genes can involve in secondary (or additional) cellular processes. Especially, in terms of *Erd1*, as the reviewer exemplified, it has been predicted to encode the membrane protein for luminal ER protein retention [1], and several studies reported that deletion of its encoding gene in yeast caused not only an ER stress response, but also the increased number of LDs and content of TAGs [2, 3]. However, despite the pleiotropic effect, there was no other reported phenotypic alterations, which was the reason why we chose the *erd1* gene for further investigation. Although the potential pleiotropic effects of some mutants cannot be completely excluded, we carefully identified ten different genes of which effects on the LD number or size have been validated by previous studies (**Table 1**). Moreover, we deleted or overexpressed each gene in a WT yeast strain and evaluated again the individual effect on the LD number or size (**Figure 3B** and **Supplementary Table 1**). For example, the *erd1* deletion in yeast was indeed effective for increasing the number of LDs, so the LD number of the WT- Δ ERD1 strain (31.5 LDs per cell) became 1.4-fold larger than that of the WT strain. Furthermore, *loa1* overexpression in this background of *erd1* deletion was observed to be highly synergistic, which was evidenced by the observation that for the LD-number strain (WT-LOA1 Δ ERD1), the number of LDs was 51.67 per cell, which was much

larger than those for WT- Δ ERD1 and WT-LOA1 (31.50 and 39.50 per cell, respectively) (**Supplementary Table 1**).

We also agree with the reviewer that mutation of some genes, such as the gene encoding Lro1 for formation of lyso-phospholipids, may alter lipid metabolism and LD lipid composition; due to different lipid compositions of LDs, migration and accumulation of hydrophobic molecules into the LDs can be changed. To exclude the possibility, we examined 10 different lipid levels of LDs for our designer yeast cells (*i.e.*, LD-size and LD-number cells) using mass spectrometry lipidomics and compared for the WT cells (see new **Figure 3D**). We note that for a professional mass spectrometry-based lipid analysis, the LD lipid compositions were measured and analyzed by Lipotype GmbH (Dresden, Germany) [4, 5]. Except for diacylglycerols (DAGs), all LD lipid compositions are quite similar among the designer yeast cells and the WT cells, and when it is considered that the DAGs are precursors of TAGs, a slightly lower level of the DAGs for the LD-size and the LD-number cells, compared for the WT cells, would be attributed to our TAG formation-promoted LD engineering. Importantly, including DAG levels, all LD lipids of the LD-number and the LD-size cells exhibited the similar mol fraction levels each other; even after the distinct LD size and number engineering, the lipidomic profiles of the engineered cells would not be significantly different, suggesting that the difference of LD lipid compositions would not be critical to impact on migration and accumulation of hydrophobic molecules into the LDs. In addition, in line with previous reports [6-8], the levels of lyso-phospholipids generated by Lro1 were too small to be detected for the LDs of LD-number, LD-size, and WT cells, so it was assumed that the level change of lyso-phospholipids by our LD engineering would not be significant to alter the LD lipid composition. Taken together, although LD lipid compositions can influence on lipid storage of LDs, our claim that the selective diffusion and consequent partitioning of lipids into LDs can be mainly attributed to the structural properties of lipids would be still valid, and the relevant discussion, along with the LD lipid composition analyses, was included in the revised manuscript.

Page 12, Line 229: **Because of our LD engineering, the total lipid and LD lipid contents of LD-size and LD-number cells slightly increased compared to those of WT cells (Figure 3D, inset), but the LD lipid compositions of the two engineered cells were quite similar each other (Figure 3D, Supplementary Table 6 and 7), thereby excluding the possibility of improved LD storage by different LD lipid compositions.**

Figure 3. Microbial LD engineering to control the size and number of cytosolic LDs. (A) LD engineering strategy for controlling the size and number of LDs. By systematically rewiring LD biogenesis and degradation pathways, we controlled the size and number of intracellular LDs. In particular, we focused on genes associated with physicochemically distinct functions, including TAG synthesis, TAG degradation, TAG precursor supply, and membrane transformation, and their functions were intentionally promoted (solid line) or suppressed (dotted line). (B) Quantification of the size and number of LDs in yeast engineered by overexpression or deletion of each of ten select genes associated with LD engineering. The graph depicts scatter boxplots of morphological changes in the LDs of individual gene-regulated yeast cells ($n = 6$). For the scatter plots, medians with interquartile ranges are displayed. The numbers in parentheses indicate the LD number in each yeast cell. (C) Frequency distribution histogram showing the size and number of LDs in the designer yeast cells ($n = 6$). From synergistic combinations of LD regulatory genes, we created two different designer yeast cells in which the LDs were much larger or in which there were more LDs (LD-size and LD-number, respectively) than the LDs of the WT cell. Differential interference contrast and confocal fluorescence microscopy images of the LDs in the designer yeast cells are shown as inset figures. Scale bar: 5 μm . (D) Comparative lipidomics of the designer yeast cells. Total lipid and LD lipid levels of LD-size and LD-number cells slightly increased compared to those of WT cells (an inset figure). When the LD-size and the LD-number cells were compared each other, their LD lipid compositions, along with the total lipid and the LD lipid levels, were quite similar each other. Relative enrichment of lipids in LDs is depicted in

mol %. Yeast cells were grown in a defined minimal medium with 2% (v/w) glucose at 30 °C for 24 h. Error bars indicate the standard deviation (* $P < 0.05$, $n = 2$). n.s. not significant, n.d. not detectable. WT, wild-type CEN.PK2-1D strain; DAG, diacylglycerol; FA, fatty acid; SE, sterol ester; PA, phosphatidate; PC, phosphatidylcholine; PE, phosphatidylethanolamine; PG, phosphatidylglycerol; PI, phosphatidylinositol; PS, phosphatidylserine; LPs, lyso-phospholipids.

[1] Hardwich *et al.* ERD1, a yeast gene required for the retention of luminal endoplasmic reticulum proteins, affects glycoprotein processing in the Golgi apparatus. 1990. *EMBO J.* 9, 623-630

[2] Fei *et al.* Conditions of endoplasmic reticulum stress stimulate lipid droplet formation in *Saccharomyces cerevisiae*. 2009. *Biochem J.* 424, 61–67

[3] Teixeira *et al.* Engineering lipid droplet assembly mechanisms for improved triacylglycerol accumulation in *Saccharomyces cerevisiae*. 2018. *FEMS Yeast Res.* doi: 10.1093/femsyr/foy060

[4] Nguyen *et al.* DGAT1-dependent lipid droplet biogenesis protects mitochondrial function during starvation-induced autophagy. 2017. *Dev Cell.* 42, 9-21

[5] John *et al.* A quantitative analysis of cellular lipid compositions during acute proteotoxic ER stress reveals specificity in the production of asymmetric lipids. 2020. *Front Cell Dev Biol.* 4, 756

[6] Ejsing *et al.* Global analysis of the yeast lipidome by quantitative shotgun mass spectrometry. 2009. *Proc Natl Acad Sci USA.* 17, 2136-2141

[7] Grillitsch *et al.* Lipid particles/droplets of the yeast *Saccharomyces cerevisiae* revisited: lipidome meets proteome. 2011. *Biochimica et Biophysica Acta.* 1811, 1165-1176

[8] Pol *et al.* Biogenesis of the multifunctional lipid droplet: lipids, proteins, and sites. 2014. *Journal of cell biology.* 204, 635-646

Major concern 2:

In Figure 4, squalene accumulation is monitored over time in various conditions and shown to increase with LD size. Zeaxanthin did not accumulate as squalene did. It is not clear whether the yeast in these experiments are converting squalene to squalene epoxide or downstream mevalonate products at the same rate. This is a key control, and needs to be investigated. Squalene processing can also be inhibited by terbinafine, and may serve as a good control for this experiment.

Response: We thank the reviewer for the insightful comment. As the reviewer pointed out, zeaxanthin did not show remarkably increased accumulation in our LD-engineered cells as squalene did. We presumed that the rigid zeaxanthin might be harbored into all kinds of membranes (*e.g.*, membrane-bound organelles and plasma membranes), including the LD

surface area, as described in the original manuscript, “(lines 265-2268) ..., we presumed that the increase in stored zeaxanthin correlated with the increase in the surface area of each LD, although membrane-bound organelles and cell membranes presumably also served as zeaxanthin-stored reservoirs.” and “(lines 303-305) We observed that the storage capability of rigid lipids (*i.e.*, zeaxanthin and β -carotene) was not remarkably enhanced in our designer yeast cells, in which the net surface area of LDs had been increased; ...” To further clarify the LD surface area effect for the storage of structurally rigid lipids, we investigated the retention of another rigid β -carotene on the LD surface as well, and the β -carotene showed the storage pattern in LDs very similar to that of zeaxanthin. Furthermore, we emphasized that using confocal microscopy, β -carotene was visually observed to be favorably adsorbed into all kinds of membranes, including the LD surface (**Figure 4C**).

However, we agree with the reviewer that it is not clear whether our lipid-producing yeast cells can efficiently convert the downstream mevalonate products; in our proof-of-concept study, the yeast cells were not extensively engineered to produce high quantities of their target lipids (squalene, zeaxanthin and β -carotene). Therefore, as the reviewer suggested, we examined the production of the three target lipids using terbinafine, an inhibitor of squalene epoxidase, which blocks the synthesis of squalene epoxide from squalene, thereby increasing squalene production. As shown below, the inhibition of squalene epoxidase by terbinafine in our engineered lipid-producing yeast cells was helpful to further increase production of all target lipids (updated **Figure 4** and **Supplementary Table 2**).

In detail, we investigated the production levels of the three target lipids in the presence of different concentrations of terbinafine (5 and 10 $\mu\text{g/mL}$) after 144-h cultivation. We note that treatment with 10 $\mu\text{g/mL}$ terbinafine was reported to be sufficient to reduce the flux of squalene processing in yeast, thereby allowing a maximal increase in squalene production [1]. As shown in updated **Figure 4B** (left), regardless of terbinafine, the titers of flexible squalene for the LD-size/SQ cells was much larger than those for the LD-number/SQ cells. In contrast, rigid zeaxanthin and β -carotene showed a larger production pattern in the LD-number cells (LD-number/ZEA and LD-number/BC cells) compared to the LD-size cells (LD-size/ZEA and LD-size/BC cells), and the overall patterns were irrelevant to use of terbinafine (**Figure 4B**, right and **Figure 4C**, left); in the presence of terbinafine (10 $\mu\text{g/mL}$), the zeaxanthin titer of the LD-number/ZEA cells was 4.62 mg/L, which was $\sim 60\%$ larger than that of the LD-size/ZEA cells (2.83 mg/L), consistent with the titer trend in the absence of terbinafine.

Using the terbinafine treatment, we succeeded in improving lipid productivity, but the lipid-dependent production pattern was conserved, indicating that our claim, the lipid-dependent positional storage in LDs, would be still valid, and to improve clarity, we newly included the production titers of target lipids in the absence and presence of terbinafine in updated **Figure 4** and **Supplementary Table 2**, and the relevant discussion was added in the revised manuscript.

Page 14, Line 274: **The lipid-dependent storage pattern in LDs was consistent even when lipid productivity was further improved. As an inhibitor of squalene epoxidase, terbinafine can be used to block the synthesis of squalene epoxide from squalene, thereby accumulating squalene³¹; when the flux of the squalene processing is reduced, the amount of downstream mevalonate products, including zeaxanthin, can be also increased. In the presence of terbinafine (10 µg/mL), we investigated the storage levels of squalene and zeaxanthin, which were subsequently compared with those in the absence of terbinafine (Figure 4B). Even though the terbinafine treatment was helpful to improve the overall lipid production for all different yeast cells, the distinctive storage patterns of squalene and zeaxanthin were not altered. Regardless of terbinafine, the titers of flexible squalene for LD-size/SQ cells were much larger than those for LD-number/SQ cells, and rigid zeaxanthin showed larger storage in LD-number/ZEA cells than in LD-size/ZEA cells.**

Page 15, Line 296: In line with our MD simulations and *in vivo* evaluation, β-carotene, a rod-like lipid, showed a greater affinity for the engineered cells with the maximal LD surface area (LD-number/BC cells) compared to the LD-enlarging cells (LD-size/BC cells) and the β-carotene-producing WT cells (WT/BC cells), **regardless of terbinafine (Figure 4C, left).**

Figure 4B and 4C

Figure 4. *In vivo* evaluation of lipid-dependent migration and storage in LDs. (A) The biosynthesis pathway of three different non-saponifiable lipids, squalene, β -carotene, and zeaxanthin. Squalene was produced by overexpressing native squalene pathway genes (*thmg1* and *erg20*). β -carotene and zeaxanthin were produced via heterologous biosynthetic pathways upon expression of genes obtained from *Pantoea agglomerans* (*crtE*, *crtB*, *crtI*, *crtY*, and *crtZ*). (B) Quantitative measurements of squalene and zeaxanthin produced by designer yeast cells (LD-number and LD-size cells). Structurally flexible squalene and rigid zeaxanthin showed a completely different storage-dependence pattern; the increase in squalene and zeaxanthin storage was significantly correlated with the increase in the total volume (size) and net surface area (number) of the LDs, respectively. (C) Localization of rigid β -carotene at the LD surface. Rod-like β -carotene showed a storage pattern in LDs very similar to that of zeaxanthin; more β -carotene was stored in cells with a net increase in LD surface area (LD-number/BC) than in cells with a greater total LD volume and wild-type β -carotene-producing cells (LD-size/BC and WT/BC cells, respectively). Using confocal microscopy, we directly observed that β -carotene (red) was favorably adsorbed into all kinds of membranes, including the LD surface (green). Yeast cells were grown in a defined minimal medium with 2% (v/w) glucose at 30 °C for 24 h and harvested before the confocal fluorescence microscopy observation. We note that β -carotene was inherently fluorescent (excitation at 450 nm and emission at 600 nm), and LDs were stained with the BODIPY fluorescent dye (excitation at 488 nm and emission at 500 nm). For production of target lipids, yeast cells were grown in a defined minimal medium with 2% (v/w) glucose in the presence (+) or absence (-) of 10 μ g/mL terbinafine at 30 °C for 144 h. The inhibition of squalene epoxidase (Erg1) by terbinafine was helpful to further increase production of squalene, zeaxanthin, and β -carotene, but it did not alter the distinctive storage patterns of the flexible and rigid lipids. All the data represent the mean of biological triplicates, and error bars indicate the standard deviation. An asterisk (*) indicates that the value is significantly different ($P < 0.05$) from that for the respective control cells. IPP, isopentenyl pyrophosphate; DMAPP, dimethylallyl diphosphate; GPP, geranyl pyrophosphate; FPP, farnesyl pyrophosphate; GGPP, geranylgeranyl pyrophosphate. Scale bar, 5 μ m.

Updated Supplementary Table 2

Target Terpene	Strain	Time (h)	Titer (mg/L)			Content (mg/g DCW)			
			Without terbinafine	5 µg/mL terbinafine	10 µg/mL terbinafine	Without terbinafine	5 µg/mL terbinafine	10 µg/mL terbinafine	
Squalene	WT/SQ	72	0.37 (±0.00)	3.21 (±0.48)	4.20 (±0.24)	0.05 (±0.00)	0.49 (±0.10)	0.67 (±0.04)	
		144	0.43 (±0.05)	4.19 (±0.03)	5.23 (±0.16)	0.06 (±0.01)	0.71 (±0.03)	0.96 (±0.08)	
	LD-number/SQ	72	1.23 (±0.06)	5.00 (±0.08)	10.52 (±0.37)	0.22 (±0.05)	0.82 (±0.00)	1.71 (±0.04)	
		144	4.57 (±0.30)	5.00 (±0.09)	13.60 (±0.10)	0.72 (±0.04)	0.90 (±0.10)	2.39 (±0.05)	
	LD-size/SQ	72	6.46 (±0.59)	7.60 (±0.82)	14.01 (±0.10)	1.03 (±0.09)	1.21 (±0.01)	2.52 (±0.17)	
		144	13.93 (±0.16)	15.43 (±0.19)	25.68 (±0.79)	1.64 (±0.02)	3.03 (±0.03)	5.02 (±0.36)	
	Zeaxanthin	WT/ZEA	72	1.26 (±0.05)	1.36 (±0.07)	1.71 (±0.01)	0.20 (±0.00)	0.22 (±0.01)	0.28 (±0.02)
			144	1.27 (±0.10)	1.58 (±0.09)	2.02 (±0.12)	0.20 (±0.01)	0.23 (±0.02)	0.33 (±0.06)
LD-number/ZE A		72	2.57 (±0.34)	3.24 (±0.07)	3.68 (±0.01)	0.46 (±0.08)	0.50 (±0.01)	0.55 (±0.01)	
		144	2.85 (±0.24)	3.57 (±0.29)	4.62 (±0.22)	0.49 (±0.04)	0.53 (±0.09)	0.68 (±0.04)	
LD-size/ZEA		72	1.91 (±0.18)	1.88 (±0.01)	2.29 (±0.02)	0.31 (±0.02)	0.31 (±0.00)	0.36 (±0.02)	
		144	2.16 (±0.13)	1.83 (±0.03)	2.83 (±0.38)	0.32 (±0.01)	0.29 (±0.00)	0.41 (±0.06)	
β-Carotene		WT/BC	72	14.04 (±0.35)	14.40 (±0.86)	16.23 (±0.05)	2.29 (±0.00)	2.46 (±0.18)	2.93 (±0.20)
			144	15.73 (±0.37)	15.45 (±0.80)	20.62 (±0.15)	2.20 (±0.02)	2.41 (±0.15)	3.58 (±0.08)
	LD-number/BC	72	14.35 (±0.79)	16.87 (±0.66)	23.80 (±0.93)	2.26 (±0.07)	2.65 (±0.10)	3.90 (±0.14)	
		144	23.28 (±12)	25.08 (±0.25)	32.65 (±0.35)	3.81 (±0.03)	4.05 (±0.13)	5.66 (±0.30)	
	LD-size/BC	72	15.70 (±0.10)	16.62 (±0.26)	17.43 (±0.13)	2.53 (±0.10)	2.48 (±0.13)	2.75 (±0.08)	
		144	16.67 (±1.08)	17.25 (±0.02)	22.11 (±0.90)	2.34 (±0.18)	2.77 (±0.05)	3.76 (±0.15)	

[1] Han *et al.* High-level recombinant production of squalene using selected *Saccharomyces cerevisiae* strains. 2018. *J Ind Microbiol Biotechnol.* 45, 239-251.

Reviewer #3

Reviewer comment:

The manuscript by Son et al. reports that the storage capacity/accumulation of lipophilic products by engineered yeast can be affected by the structural properties of the lipophilic products and the size/surface area of the lipid droplets in the yeast cells. The authors used MD simulations to demonstrate the differential deposition locations of squalene and beta-carotene in lipid droplets. The worm-like flexible structure of squalene enabled deeper penetration into the core of lipid droplets. In contrast, the rod-like rigid structure of zeaxanthin and beta-carotene led to the trapping of the zeaxanthin and beta-carotene on the surface of lipid droplets. To confirm the differential localization of squalene, beta-carotene, and zeaxanthin *in vivo*, the authors constructed *Saccharomyces cerevisiae* mutant strains containing varied numbers and sizes of lipid droplets. Two mutant *S. cerevisiae* strains with increased sizes (LD-size) and numbers (LD-number) of lipid droplets were selected and used to examine squalene, beta-carotene, and zeaxanthin production. As predicted by the MD simulations, the LD-size strain was superior to the LD-number strain for squalene accumulation. On the contrary, the LD-number strain was slightly better than the LD-size strain for accumulating beta-carotene and squalene.

The storage capacity (mainly in terms of lipid content) of hydrophobic molecules in engineered yeast has been pointed as a potential limiting factor for industrial production. However, the size and number of lipid droplets have not been considered for the enhanced production of hydrophobic molecules. Therefore, the main idea of the study is new and interesting but *in vivo* validation will need more rigorous experiments to support the main argument of the study.

Overall response: We sincerely thank you for appreciating the novelty and the significance of our work. Due to your encouraging comments and insightful suggestions, we strongly believe that the manuscript would become further strengthened with additional *in vivo* experiments. We tried our best to answer all your questions in the point-by-point responses as listed below. The changes in the revised manuscript were highlighted in red accordingly.

Reviewer comment 1:

The authors did not measure the total lipid contents of their engineered yeast strains. While the number and size of lipid droplets are the main targets of their engineering, total lipid content has been reported as a primary differentiator for the enhanced storage of hydrophobic molecules. As such, the authors will need to show the total lipid contents of LD-size and LD-number strains are equivalent. Then the higher accumulation of squalene in the LD-size than the LD-number strain can be attributed to the structural properties of squalene and LDs. This reviewer suggests including the lipid contents of the engineered strains shown in Figure B.

Response: We thank the reviewer for the insightful comment. As the reviewer pointed out, total lipid contents of cells can affect accumulation of hydrophobic molecules; a previous study has reported that the elevated triacylglycerol (TAG) levels of LDs improved lipophilic lycopene accumulation in *S. cerevisiae* [1]. Hence, to evaluate our claim, “the selective diffusion and consequent accumulation of lipids into LDs can be attributed to the structural properties of lipids,” it should be validated that the total lipid contents of LD-size and LD-number strains are equivalent.

According to the reviewer’s suggestion, we newly examined the total lipid contents in our designer yeast cells (LD-number and LD-size cells) using mass spectrometry lipidomics (see new **Figure 3D**); for a professional mass spectrometry-based lipid analysis, the total lipid contents were measured and analyzed by Lipotype GmbH (Dresden, Germany) [2, 3]. As shown in new **Figure 3D**, the total lipid contents of the LD-number and LD-size cells were measured to be 10,488 and 9,385 pmol/g DCW, respectively, exhibiting the similar values (~10% difference only). Moreover, it was confirmed that both cells were very similar in total LD lipid compositions as well. From those observations, we could exclude the possibility of different lipid contents in increasing the lipid storage in LDs, thereby further supporting our claim, lipid-dependent positional storage in the LDs. Thus, along with the LD lipid composition analyses, the relevant discussion was included in the revised manuscript.

Page 12, Line 229: **Because of our LD engineering, the total lipid and LD lipid contents of LD-size and LD-number cells slightly increased compared to those of WT cells (Figure 3D, inset), but the LD lipid compositions of the two engineered cells were quite similar each other (Figure 3D, Supplementary Table 6 and 7), thereby excluding the possibility of improved LD storage by different LD lipid compositions.**

Figure 3. Microbial LD engineering to control the size and number of cytosolic LDs.

... **(D)** Comparative lipidomics of the designer yeast cells. Total lipid and LD lipid levels of LD-size and LD-number cells slightly increased compared to those of WT cells (an inset figure). When the LD-size and the LD-number cells were compared each other, their LD lipid compositions, along with the total lipid and the LD lipid levels, were quite similar each other. Relative enrichment of lipids in LDs is depicted in mol %. Yeast cells were grown in a defined minimal medium with 2% (v/w) glucose at 30 °C for 24 h. Error bars indicate the standard deviation (* $P < 0.05$, $n = 2$). n.s. not significant, n.d. not detectable. WT, wild-type CEN.PK2-1D strain; DAG, diacylglycerol; FA, fatty acid; SE, sterol ester; PA, phosphatidate; PC, phosphatidylcholine; PE, phosphatidylethanolamine; PG, phosphatidylglycerol; PI, phosphatidylinositol; PS, phosphatidylserine; LPs, lyso-phospholipids.

[1] Ma *et al.* Lipid engineering combined with systematic metabolic engineering of *Saccharomyces cerevisiae* for high-yield production of lycopene. 2019. *Metab Eng.* 52, 134–142

[2] Nguyen *et al.* DGAT1-dependent lipid droplet biogenesis protects mitochondrial function during starvation-induced autophagy. 2017. *Dev Cell.* 42, 9-21

[3] John *et al.* A quantitative analysis of cellular lipid compositions during acute proteotoxic ER stress reveals specificity in the production of asymmetric lipids. 2020. *Front Cell Dev Biol.* 4, 756

Reviewer comment 2:

P-values from statistical testing of the product concentrations in Figures 4B and 4C need to be presented.

Response: We thank the reviewer for the important comment. We performed the statistical analysis of the product concentrations and amended **Figure 4B** and **4C** to include *P*-values. As shown below, the product concentrations (squalene, zeaxanthin and β -carotene) were statistically significant ($P < 0.05$) in designer yeast cells (LD-number and LD-size cells). The details are in the figure legend, “(Line 694) An asterisk (*) indicates that the value is significantly different ($P < 0.05$) from that for the respective control cells.”

Figure 4B and 4C

Figure 4. *In vivo* evaluation of lipid-dependent migration and storage in LDs. ... (B) Quantitative measurements of squalene and zeaxanthin produced by designer yeast cells (LD-number and LD-size cells). Structurally flexible squalene and rigid zeaxanthin showed a completely different storage-dependence pattern; the increase in squalene and zeaxanthin storage was significantly correlated with the increase in the total volume (size) and net surface area (number) of the LDs, respectively. **(C)** Localization of rigid β -carotene at the LD surface. Rod-like β -carotene showed a storage pattern in LDs very similar to that of zeaxanthin; more β -carotene was stored in cells with a net increase in LD surface area (LD-number/BC) than in cells with a greater total LD volume and wild-type β -carotene-producing cells (LD-size/BC and WT/BC cells, respectively). Using confocal microscopy, we directly observed that β -carotene (red) was favorably adsorbed into all kinds of membranes, including the LD surface (green). Yeast cells were grown in a defined minimal medium with 2% (v/w) glucose at 30 °C for 24 h and harvested before the confocal fluorescence microscopy observation. We note that β -carotene was inherently fluorescent (excitation at 450 nm and emission at 600 nm), and LDs were stained with the BODIPY fluorescent dye (excitation at 488 nm and emission at 500 nm). For production of target lipids, yeast cells were grown in a defined minimal medium with 2% (v/w) glucose in the presence (+) or absence (-) of 10 μ g/mL terbinafine at 30 °C for 144 h. The inhibition of squalene epoxidase (Erg1) by terbinafine was helpful to further increase production of squalene, zeaxanthin, and β -carotene, but it did not alter the distinctive storage patterns of the flexible and rigid lipids. All the data represent the mean of biological triplicates, and error bars indicate the standard deviation. An asterisk (*) indicates that the value is significantly different ($P < 0.05$) from that for the respective control cells. IPP, isopentenyl pyrophosphate; DMAPP, dimethylallyl diphosphate; GPP, geranyl pyrophosphate; FPP, farnesyl pyrophosphate; GGPP, geranylgeranyl pyrophosphate. Scale bar, 5 μ m.

Reviewer comment 3:

The presented squalene concentrations by WT/SQ, LD-size/SQ, and LD-number/SQ in Figure 4B were much lower than the reported concentrations (30-150 mg/L) in other studies. As such, the statement in the abstract, “we observed that intracellular storage of squalene was dramatically increased, by ~3100%, with LD volume expansion” might be exaggerating. Also, the Production comparison in the Supplementary Table 3 might not be correct.

Response: We greatly thank the reviewer for bringing this information to our attention. As the reviewer pointed out, some studies achieved large squalene productions of 30-150 mg/L in *S. cerevisiae*, mainly by overexpression of multiple key genes that are associated with the squalene biosynthesis (e.g., tHMG1, IDI1, ERG20 and ERG9 genes), and multicopy-based expression and downregulation of competitive pathway genes were also employed for the production improvement [1-4]. Unlike them, this proof-of-concept study, which focused on precise LD remodeling, overexpressed only two rate-limiting enzymes, tHMG1 and ERG20, for the squalene production using the single-copy plasmid, so squalene production by our squalene-producing strains (WT/SQ, LD-size/SQ, and LD-number/SQ strains) was lower than those of the previous reported studies; thus, for appropriate storage comparison, we compared our engineered strains with their respective control strains.

However, we agree with the reviewer that in this study, describing the intracellular storage of squalene as the dramatic increase would be misleading. Therefore, we toned down the sentence: (lines 43-45) **we observed that intracellular storage of squalene was significantly increased with LD volume expansion, but that of zeaxanthin was enhanced through LD surface broadening.** In addition, “Production comparison” in **Supplementary Table 3** is defined as the fold increase of terpene production by each LD engineering strategy. In other words, when one strategy enables to increase production of its target terpene, the production comparison is referred to “the ratio of the amount of production by the engineered cells for each approach to that by the non-engineered cells”, which was stated as a remark in **Supplementary Table 3**.

[1] Rasool *et al.* Overproduction of squalene synergistically downregulates ethanol production in *Saccharomyces cerevisiae*. 2016. *Chemical Engineering Science*. 152, 370-380

[2] Paramasivan *et al.* Regeneration of NADPH coupled with HMG-CoA reductase Activity increases squalene synthesis in *Saccharomyces cerevisiae*. 2017. *J Agric Food Chem*. 65, 8162-8170

[3] Han *et al.* High-level recombinant production of squalene using selected *Saccharomyces cerevisiae* strains. 2018. *J Ind Microbiol Biotechnol*. 34, 239-251

[4] Gohil *et al.* Engineering strategies in microorganisms for the enhanced production of squalene: Advances, challenges and opportunities. 2019. *Front Bioeng Biotechnol*. 7, 50

Reviewer comment 4:

The authors are using only titers of squalene, beta-carotene, and zeaxanthin (mg/L) for comparing the engineered strains. As the lipophilic products are intracellular products, the specific contents (mg/g cell) need to be also measured and considered for the comparison as in

the previous studies.

Response: We thank the reviewer for the insightful comment. We strongly agree with the reviewer that as squalene, β -carotene, and zeaxanthin are intracellularly-accumulated lipophilic products, measuring the product content per cell, *i.e.*, the specific content, would be helpful to compare the amounts of accumulated products within cells. As suggested, we newly included the specific contents (mg/g DCW) of squalene, β -carotene, and zeaxanthin in the updated **Figure 4B** and **4C**; the specific contents and the titers of the lipophilic products displayed very similar patterns each other.

Reviewer comment 5:

The authors will need to describe when the cells were harvested for obtaining microscopic images in Figure 4C and how beta-carotene was visualized.

Response: We are sorry for the unclear explanation. With regard to **Figure 4C**, we harvested the β -carotene-producing cells after 24 h culture in a defined minimal medium supplemented with 2% (w/v) glucose at 30 °C, as stated on methods section of the original manuscript (lines 384-387). For the confocal fluorescence microscopy observation, β -carotene was inherently fluorescent (excitation at 450 nm and emission at 600 nm), but LDs were stained with the BODIPY fluorescent dye (excitation at 488 nm and emission at 500 nm) [1-3]. We have included the experimental details in the legend of **Figure 4** to support readers' better understanding.

Page 37, Line 684: Yeast cells were grown in a defined minimal medium with 2% (v/w) glucose at 30 °C for 24 h and harvested before the confocal fluorescence microscopy observation. We note that β -carotene was inherently fluorescent (excitation at 450 nm and emission at 600 nm), and LDs were stained with the BODIPY fluorescent dye (excitation at 488 nm and emission at 500 nm).

[1] Pawlak, K., Skrzypczak, A., & Bialek-Bylka, G. E. (2013). Fluorescence emission characterization of all-trans and 15-cis- β -carotene in imidazolium based room temperature ionic liquids. *Dyes and Pigments*, 99(1), 168-175.

[2] Lee, J., Song, J., Lee, D., & Pang, Y. (2019). Metal-enhanced fluorescence and excited state dynamics of carotenoids in thin polymer films. *Scientific reports*, 9(1), 1-12.

[3] Domenici, V., Ancora, D., Cifelli, M., Serani, A., Veracini, C. A., & Zandomeneghi, M. (2014). Extraction of pigment information from near-UV vis absorption spectra of extra virgin olive oils. *Journal of agricultural and food chemistry*, 62(38), 9317-9325.

Reviewers' Comments:

Reviewer #1:

Remarks to the Author:

The authors have addressed all my comments and that of other reviewers as per the detailed response submitted with the revised manuscript. The manuscript is significantly stronger and I recommend it for publication.

Reviewer #2:

Remarks to the Author:

The revised manuscript has addressed the majority of my concerns. My major concern was the lack of analysis of potentially pleiotrophic effects some yeast mutants may have on lipid biology. The revision has done several text changes, and a few new experiments, to help address this. In particular, the new experiments with terbinafine largely address the concern about squalene production and flux through the mevalonate pathway.

Reviewer #3:

Remarks to the Author:

The authors made revisions appropriately. However, the lipid contents (10,488 and 9,385 pmol/g DCW) of LD-number and LD-size are not consistent with other contents in the manuscript. For instance, squalene and zeaxanthin contents were shown in mg/g DCW. Please show the total lipid contents in mg/g DCW.

With an average molecular weight of 867 g/mol of TAG, the 10,488 pmol is estimated to ~9 micro g (0.009 mg) which is multiple orders of magnitude lower than the typical lipid contents of yeast cells. This reviewer thinks that an experimental validation (by traditional methods) of total lipid contents measured by Lipotype GmbH will be necessary.

Point-by-Point Response

Revised Title: Chain flexibility of medicinal lipids determines their selective partitioning into lipid droplets

Corresponding Authors: Ju Young Lee & Seung Soo Oh & Chang Yun Son

Authors: So-Hee Son et al.

Reviewer #1

Reviewer overall comment:

The authors have addressed all my comments and that of other reviewers as per the detailed response submitted with the revised manuscript. The manuscript is significantly stronger and I recommend it for publication.

Response: We would like to thank the reviewer for your helpful comments, and for your time and effort in reviewing our manuscript.

Reviewer #2

Reviewer overall comment:

The revised manuscript has addressed the majority of my concerns. My major concern was the lack of analysis of potentially pleiotropic effects some yeast mutants may have on lipid biology. The revision has done several text changes, and a few new experiments, to help address this. In particular, the new experiments with terbinafine largely address the concern about squalene production and flux through the mevalonate pathway.

Response: We would like to thank the reviewer for useful comments and suggestions to improve our manuscript.

Reviewer #3

Reviewer comment:

The authors made revisions appropriately. However, the lipid contents (10,488 and 9,385 pmol/g DCW) of LD-number and LD-size are not consistent with other contents in the manuscript. For instance, squalene and zeaxanthin contents were shown in mg/g DCW. Please show the total lipid contents in mg/g DCW.

With an average molecular weight of 867 g/mol of TAG, the 10,488 pmol is estimated to ~9 micro g (0.009 mg) which is multiple orders of magnitude lower than the typical lipid contents of yeast cells. This reviewer thinks that an experimental validation (by traditional methods) of total lipid contents measured by Lipotype GmbH will be necessary.

Response: We thank the reviewer for the helpful comment. In response to the reviewer's suggestion, we have amended the lipid contents of LD-number and LD-size cells with the same unit ($\mu\text{g/g DCW}$) as that of squalene and zeaxanthin contents in the updated **Figure 3D**.

As suggested, we newly measured the total lipid contents in our designer yeast cells (LD-number and LD-size cells), according to a traditional method using gravimetric determination [1]; as the reviewer expected, the total lipid contents of the LD-number and LD-size cells were measured to be approximately 6.26 and 6.06 $\mu\text{g/g DCW}$, respectively (see the point-by-point response Figure 1 as shown below), exhibiting the very similar values measured by Lipotype GmbH (7.73 and 6.90 $\mu\text{g/g DCW}$ of LD-number and LD-size, respectively).

With regard to the lower lipid contents of our yeast cells that reviewer pointed out, a previous study has reported that the lipid level of yeast cells would be very variable and depend strongly on growth conditions, like temperature, carbon source, overexpression of different proteins and growth phase (log-phase, stationary-phase, etc.) [2]. Similarly, Lipotype GmbH answered that the various culture conditions and growing phases could influence the lipidome of yeast cells, meaning that as yeast cells and growth conditions are different between our results and the others', their direct and fair comparisons may not work well.

Updated Figure 3D.

Point-by-point response Figure 1. The total lipid contents in our designer yeast cells (LD-number and LD-size cells) was gravimetrically determined by solvent extractions using the traditional method. The total lipid contents measured by the traditional method and Lipotype GmbH were very similar each other.

[1] Liang *et al.* Complete and efficient conversion of plant cell wall hemicellulose into high-value bioproducts by engineered yeast. 2021. *Nat Commun.* 12, 4975.

[2] Christian Klose *et al.* Flexibility of a Eukaryotic Lipidome – Insights from Yeast Lipidomics. 2012. *PLoS One*. 7:e35063.

Reviewers' Comments:

Reviewer #3:

Remarks to the Author:

Thanks for converting the lipid contents from pmol/DCW to ug/DCW.

Many studies have determined the cellular composition of *S. cerevisiae*. There are some variations due to different substrates and culture conditions, as the authors mentioned. However, the average lipid content of *S. cerevisiae* is 70 mg total lipids/g dried cell weight (DCW) = 7 % (de Jong-Gubbels, Yeast 11:407) and 7 mg TAG/g DCW (Ferreira, Metabolic Engineering Communications 6:22). As lipids are parts of essential structures (membranes) of cells, there are minimum amounts to be present in cells.

The lipid content (6.26 and 6.06 ug/g DCW) reported by the authors are three-order magnitudes lower than typical lipid contents (3-7 mg/g DCW) of yeast cells.

Let's do this calculation: 10,488 pmol/g DCW (reported by the authors) means that there are 6.29×10^{15} molecules of lipids in g DCW of the LD cells. In Figure 4, the LD cells can accumulate up to 2 mg squalene/g dcw, meaning that 4.8×10^{17} molecules of squalene in the LD cells. This does not make sense at all. The numbers of squalene molecules are 100 times higher than that of the TOTAL lipid molecules which are serving as a storage place (as shown in Figure 2A). With these numbers of squalene and zeaxanthin and lipid molecules, Figure A and B need to be re-drawn, and the simulation results cannot be justified.

The reported contents of their products are in the range from 0.5 to 4 mg/g DCW. It is simply impossible to store such large amounts in 0.006-7 mg lipid/g DCW.

This reviewer strongly believes that there was an error in measuring lipid contents in the cells. Please consider fixing the error.

Point-by-Point Response

Revised Title: Chain flexibility of medicinal lipids determines their selective partitioning into lipid droplets

Corresponding Authors: Ju Young Lee & Seung Soo Oh & Chang Yun Son

Authors: So-Hee Son et al.

Reviewer #3

Reviewer comment:

Thanks for converting the lipid contents from pmol/DCW to ug/DCW.

Many studies have determined the cellular composition of *S. cerevisiae*. There are some variations due to different substrates and culture conditions, as the authors mentioned. However, the average lipid content of *S. cerevisiae* is 70 mg total lipids/g dried cell weight (DCW) = 7 % (de Jong-Gubbels, Yeast 11:407) and 7 mg TAG/g DCW (Ferreira, Metabolic Engineering Communications 6:22). As lipids are parts of essential structures (membranes) of cells, there are minimum amounts to be present in cells.

The lipid content (6.26 and 6.06 ug/g DCW) reported by the authors are three-order magnitudes lower than typical lipid contents (3-7 mg/g DCW) of yeast cells.

Let's do this calculation: 10,488 pmol/g DCW (reported by the authors) means that there are 6.29×10^{15} molecules of lipids in g DCW of the LD cells. In Figure 4, the LD cells can accumulate up to 2 mg squalene/g dcw, meaning that 4.8×10^{17} molecules of squalene in the LD cells. This does not make sense at all. The numbers of squalene molecules are 100 times higher than that of the TOTAL lipid molecules which are serving as a storage place (as shown in Figure 2A). With these numbers of squalene and zeaxanthin and lipid molecules, Figure A and B need to be re-drawn, and the simulation results cannot be justified.

The reported contents of their products are in the range from 0.5 to 4 mg/g DCW. It is simply impossible to store such large amounts in 0.006-7 mg lipid/g DCW.

This reviewer strongly believes that there was an error in measuring lipid contents in the cells. Please consider fixing the error.

Response: We sincerely appreciate the reviewer for the constructive and helpful comments. We strongly agree with the reviewer that yeast cells should produce minimum levels of lipids for their living, and based on the reviewer's keen observation, there should have been a technical error in measuring lipid contents in the cells in the previous revised manuscript. At this time, to newly determine the total lipids from our designer yeast cells (LD-number and LD-size cells), we relied on the different lipid extraction protocol that enabled complete recovery of total lipids from cells by applying a large volume of solvent [1-3].

When the total lipid contents were newly measured after 144 h of cultivation, those of the LD-number and LD-size cells were approximately 54.607 and 49.176 mg/g DCW, respectively, quite similar with the average lipid content of *S. cerevisiae* (70 mg/g DCW); as the cultivation time increased, the total lipid contents also increased, but still, there was no significant difference in the total lipid contents among the wild-type, the LD-size, and the LD-number cells. It is well-known that the optimal cell-to-extraction solvent ratio is vital for complete lipid extraction [4], and presumably, during our previous lipid content measurements, the desired lipids could be incompletely extracted from yeast cells or insufficiently dissolved in solvent. We apologize the technical error and now believe it is fixed well. Moreover, we included the newly obtained results about the total lipid contents of our designer cells for 144 h of cultivation in the Supplementary Table 6, and we corrected the Figure 3D (see below).

[1] Liang *et al.* Complete and efficient conversion of plant cell wall hemicellulose into high-value bioproducts by engineered yeast. 2021. *Nat Commun.* 12, 4975.

[2] Yunfeng Ding *et al.* Isolating lipid droplets from multiple species. 2013. *Nat Protoc.* 1, 43-51.

[3] Zhiwei Zhu *et al.* Dynamics of the lipid droplet proteome of the Oleaginous yeast *rhodosporidium toruloides*. 2015. *Eukaryot Cell.* 3, 252-264.

[4] Candice Z Ulmer *et al.* Optimization of Folch, Bligh-Dyer, and Matyash sample-to-extraction solvent ratios for human plasma-based lipidomics studies. 2018. *Anal Chim Acta.* 1037, 351-357.

Updated Supplementary Table 6. Total lipid levels of whole cells and LDs quantified in LD size and LD-number strains.

Strains		Total lipids (mg/g DCW)					
		24 h	48 h	72 h	96 h	120 h	144 h
Whole Cells	WT	2.774 (±0.057)	3.325 (±1.059)	4.053 (±0.092)	33.167 (±2.256)	35.131 (±0.098)	44.827 (±1.563)
	LD-size	3.465 (±0.001)	4.534 (±0.170)	4.179 (±1.017)	38.487 (±2.029)	43.936 (±0.345)	49.176 (±8.695)
	LD-number	3.419 (±0.168)	4.772 (±0.962)	4.924 (±0.481)	42.778 (±1.386)	43.350 (±0.860)	54.607 (±5.853)
LDs	WT	0.024 (±0.000)	0.052 (±0.000)	0.092 (±0.003)	0.533 (±0.001)	0.554 (±0.015)	0.618 (±0.005)
	LD-size	0.052 (±0.001)	0.092 (±0.011)	0.134 (±0.009)	0.675 (±0.009)	0.670 (±0.017)	0.777 (±0.009)
	LD-number	0.054 (±0.001)	0.108 (±0.001)	0.149 (±0.004)	0.708 (±0.006)	0.724 (±0.010)	0.832 (±0.018)

Updated Figure 3D.

Reviewers' Comments:

Reviewer #3:

Remarks to the Author:

The authors made appropriate revisions.

Point-by-Point Response

Revised Title: Chain flexibility of medicinal lipids determines their selective partitioning into lipid droplets

Corresponding Authors: Ju Young Lee & Seung Soo Oh & Chang Yun Son

Authors: So-Hee Son et al.

We note that only the single comment below was received in the final revision. To show the history of previous revisions, we have included point-by-point response to the reviewer comments in previous revisions in next pages.

Reviewer #3

Reviewer comment:

The authors made appropriate revisions.

Response:

We sincerely thank the reviewer for the constructive and helpful comments and exceptionally useful suggestions that helped us to further strengthen the manuscript.